# Predicting the evolution of the Lassa virus endemic area and population at risk over the next decades

Raphaëlle Klitting [1] ✉, Liana E. Kafetzopoulou [2,3], Wim Thiery [4], Gytis Dudas [5], Sophie Gryseels[6,7], Anjali Kotamarthi[1], Bram Vrancken [2], Karthik Gangavarapu[1], Mambu Momoh[8,9], John Demby Sandi[9], Augustine Goba[9], Foday Alhasan[9], Donald S. Grant [9,10], Sylvanus Okogbenin[11,12], Ephraim Ogbaini-Emovo[11], Robert F. Garry [13,14,15], Allison R. Smither[13], Mark Zeller[1], Matthias G. Pauthner [1], Michelle McGraw[1], Laura D. Hughes [16], Sophie Duraffour[3,17], Stephan Günther [3,17], Marc A. Suchard [18,19,20], Philippe Lemey [2], Kristian G. Andersen[1,21] & Simon Dellicour [2,22] ✉

Lassa fever is a severe viral hemorrhagic fever caused by a zoonotic virus that repeatedly spills over to humans from its rodent reservoirs. It is currently not known how climate and land use changes could affect the endemic area of this virus, currently limited to parts of West Africa. By exploring the environmental data associated with virus occurrence using ecological niche modelling, we show how temperature, precipitation and the presence of pastures determine ecological suitability for virus circulation. Based on projections of climate, land use, and population changes, we find that regions in Central and East Africa will likely become suitable for Lassa virus over the next decades and estimate that the total population living in ecological conditions that are suitable for Lassa virus circulation may drastically increase by 2070. By analysing geotagged viral genomes using spatially-explicit phylogeography and simulating virus dispersal, we find that in the event of Lassa virus being introduced into a new suitable region, its spread might remain spatially limited over the first decades.

Along with other viral infections that have gained prominence in recent years[1–3], Lassa fever (Lassa) is listed by the World Health Organization (WHO) as one of the diseases that pose the greatest public health risk[4,5]. Lassa is a viral hemorrhagic fever with variable but generally high case fatality rates[6] for which efficacious countermeasures are lacking[7,8]. There is currently no vaccine approved to prevent Lassa. Although several candidates have shown promising results during preclinical studies, only one (INO-4500) has progressed to clinical trials (now in phase 1B, NCT03805984)[9,10]. Regarding Lassa treatment, the only antiviral drug available is the nucleoside analog ribavirin[11], which is often ineffective[7]. To date, Lassa cases have mostly been reported in West Africa including Guinea, Liberia, Nigeria, and Sierra Leone. While these countries seem to constitute most endemic hotspots[12], local under-reporting could potentially bias this overall picture. Nigeria, in particular, has witnessed a significant increase in incidence in recent years, and confirmed more than a thousand cases in 2020[13]. Increasingly, neighbouring countries, including Benin, Ghana, Ivory Coast, Mali, and Togo, have also been reporting infections[14–17], suggesting that the true Lassa range may span a sizable part of West Africa.

Lassa is caused by Lassa virus[18], a member of the *Arenaviridae* family (genus *Mammarenavirus*). Human infections are generally thought to occur through direct contact or exposure to the excreta of infected *Mastomys natalensis*[19–23], although the main transmission mechanism remains to be formally established. *M. natalensis* rodents often live in close contact with human communities[24] and are regarded

---

as the primary reservoir for the virus[25]. Humans likely contribute little to virus transmission and are considered dead-end hosts, based on studies of rodent biology[18], ecology[11,26–28], transmission dynamics[22,29], and viral genomes[30–32]. While the virus can only spread where its reservoir is present, the range of *M. natalensis* extends beyond that of Lassa virus, spanning most of sub-Saharan Africa[33,34]. The factors underlying this difference in range between the virus and its reservoir have led to long-standing questions about suitability and may be multifactorial: Lassa virus might exclusively circulate within one of the six *M. natalensis* phylogroups or subspecies, namely A-I[35], which is only found in West Africa[35–37]; other viruses present in *M. natalensis* populations may prevent Lassa virus circulation through competition[38,39]; closely related mammarenaviruses inducing cross-reactive immunity[40–42] may prevent Lassa virus infection; and finally, Lassa virus prevalence may be influenced by different environmental determinants than its reservoir, as in the case of Sin Nombre virus[43]. For this other rodent-borne virus, environmental conditions can impact the abundance of the host, driving the population density of the reservoir below the threshold needed for virus maintenance[43–45].

Ecological niche modelling studies have identified−although not always agreed on − environmental factors that correlate with the occurrence of Lassa virus infections in rodent and human hosts[34,46–49]. Additional biological and socio-ecological factors may further influence spill-over dynamics, as suggested by mechanistic modelling investigations[46,49,50]. Most of these studies have mapped the current risk for Lassa virus infection in West Africa, identifying risk areas across much of this region[34,46–49]. Like the rest of the world, African countries will increasingly be affected by climate change, with warming temperatures and more extreme, yet rarer, precipitation[51–53]. These changes, combined with an increasing pressure on land resources due to a considerable projected human population expansion, are expected to result in important transformations of land use throughout Africa[54–56]. Previously, Redding and colleagues showed that spill-over events would at least double by 2070 within the Lassa-endemic western African region due to climate change, human population growth, and to a smaller extent, land use changes[46]. It is not known, however, how these environmental changes may affect the distribution of the virus itself[57].

In this work, we add to previous modelling studies by combining ecological niche modelling and phylogeographic analyses to investigate how the endemic range of Lassa virus may evolve in the next five decades in response to climate change, human population growth, and land use changes. We start with using ecological niche modelling to assess whether regions outside the current endemic range may become suitable for Lassa virus due to changes in environmental conditions. To identify factors driving suitability for virus circulation, we analyse the environmental data associated with the occurrence of Lassa virus for a set of putative explanatory factors. We find that annual mean temperature, annual precipitation, and the presence of pastures are the main factors determining ecological suitability for Lassa virus circulation. Using projections of climate, land use, and population up to 2070, we estimate future ecological suitability and show that within decades, the range suitable for Lassa virus may extend well beyond West Africa. Using population projections, we then estimate that the extended part of the suitable range will be home to an increasingly large number of people in the next decades. We hypothesise that if Lassa virus is introduced into a new suitable region, it could in theory spread there because its reservoir host is present. This process, however, might take time. To gain insights into how fast the virus may be able to spread through a suitable environment, we subsequently perform spatially-explicit phylogeographic analyses. Using geotagged viral genomes, we show that over the first decades following a successful introduction into a new region its propagation could remain spatially limited, unless the virus spreads significantly faster than in current endemic areas. By combining ecological niche modelling with spatially-explicit phylogeography, our study showcases how climate and land use change may transform the future risk of Lassa in Africa.

## Results

### Main determinants of ecological suitability for Lassa virus

To identify factors that determine ecological suitability for Lassa virus and *M. natalensis*, we built ecological niche models, considering temperature, precipitation, seven types of land cover, and human population as potential determinants. Using a boosted regression trees[58] (BRT) method, we searched for associations between known occurrences of the virus and its reservoir and the environmental conditions at those sites. As inputs for our models, we used occurrence records collated from online databases and the literature and environmental data obtained from the Inter-Sectoral Impact Model Intercomparison Project phase 2b (ISIMIP2b)[59]. To assess how each factor contributed to our models, we calculated their relative importance (RI). In the case of BRT models, RI is evaluated based on the number of times the factor is selected for splitting a tree, weighted by the squared improvement to the model as a result of each split, averaged over all trees[59].

We found that for Lassa virus, three main factors contributed to the models: temperature (RI = 20.7%), precipitation (RI = 24.5%), pastures and rangeland coverage (RI = 25.3%; Fig. 1). For *M. natalensis*, we found that precipitation was the main contributor (RI = 50.4%; Fig. 1). These findings suggest that temperature, precipitation and the presence of pastures/rangeland may be the main factors influencing ecological suitability for Lassa virus, but not its reservoir species, *M. natalensis*, for which only precipitation appears to be critical.

To assess the relationship between each of our environmental factors and ecological suitability, we plotted response curves, which show how ecological suitability varies with one specific factor, while all others are kept constant at their mean. Ecological suitability values vary between 0 (unsuitable conditions) and 1 (highly suitable conditions). We found that temperatures below 25 °C or values of pastures and rangeland coverage below 20% seem unsuitable for the virus (ecological suitability -0; Fig. 1) but still appear relatively suitable (ecological suitability >0.4) for its reservoir species. These results indicate that even if *M. natalensis* may be found in areas with mean daily temperatures below 25 °C and limited pastures and rangeland coverage, Lassa virus is not likely to be present.

### Likely expansion of the range suitable for Lassa virus

Our ecological niche modelling analyses showed that temperature, precipitation, and pastures/rangeland coverage are the main factors influencing ecological suitability for Lassa virus circulation. Due to climate change and increasing human pressure on land resources caused by population growth, these variables are expected to change in the next decades[54–56]. With these expected transformations, the overall area suitable for Lassa virus − also called the ecological niche of the virus[60]− will likely undergo substantial changes and expand. To investigate this, we used climate and land cover projections from the year 2030 to 2070 to estimate the future ecological suitability for the virus across Africa. We found that the ecological niche of Lassa virus will likely expand as new regions become suitable, notably in Central and East Africa.

We used our ecological niche models to identify the areas suitable for Lassa virus throughout Africa, based either on current or projected values of temperature, precipitation, land cover, and human population from the ISIMIP2b[59]. We considered environmental values projected at three-time points (2030, 2050, and 2070) according to three climate scenarios: representative concentration pathways (RCPs) 2.6, 6.0, and 8.5 − which describe the evolution of global warming depending on different trajectories of greenhouse gases atmospheric concentrations[61]. For the present-day situation, our ecological niche maps for Lassa virus and *M. natalensis* were in agreement with previous estimates[34,48], showing Lassa virus suitability across West Africa,

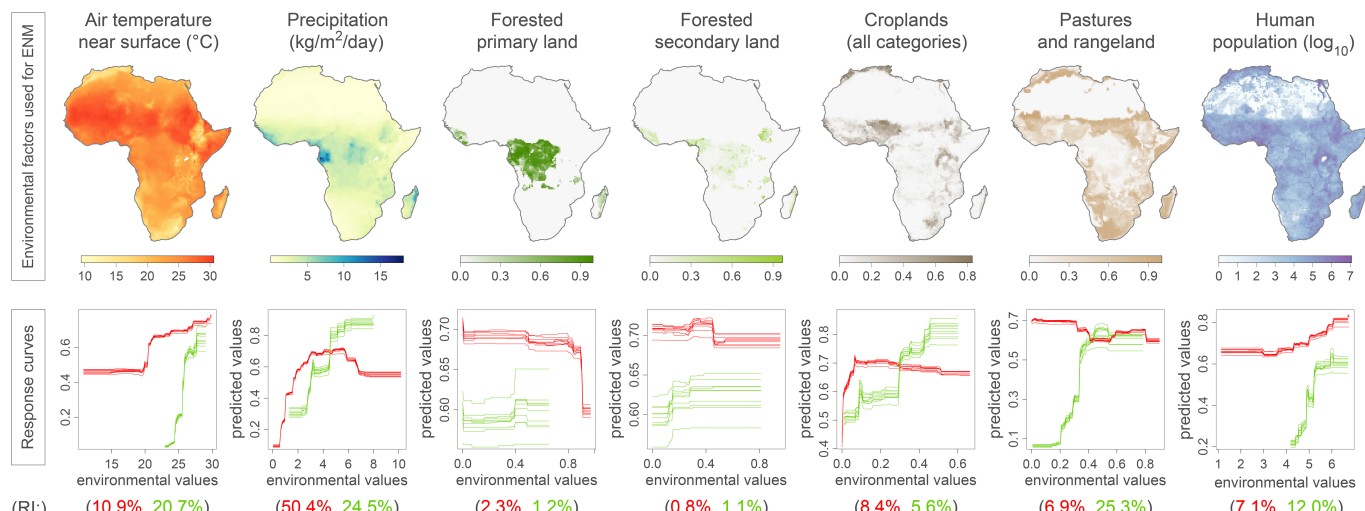

**Fig. 1 | Environmental factors included in the ecological niche modelling (ENM) analyses of *Mastomys natalensis* and Lassa virus, and their corresponding ENM response curves.** Response curves and relative importance (RI) obtained for the ENM analyses of *M. natalensis* and Lassa virus are coloured in red and green, respectively. The ten response curves reported for each ENM analysis correspond to ten independent repetitions of the boosted regression trees (BRT) analysis.

These response curves indicate the relationship between the environmental values and the response (i.e., the ecological suitability of *M. natalensis* or Lassa virus). In addition to the seven environmental factors displayed in this figure, two additional factors were also included in the ENM analyses, the non-forested primary land, and non-forested secondary land.

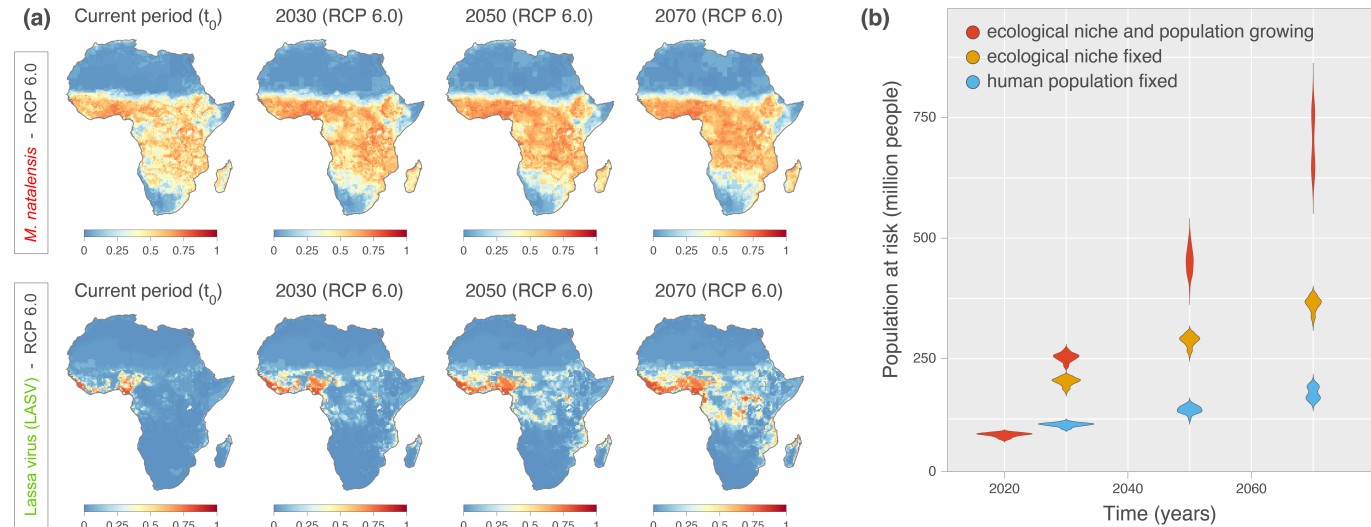

**Fig. 2 | Projected ecological niche suitability of *Mastomys natalensis* and Lassa virus, as well as human population at risk of exposure to Lassa virus. a** Projected ecological niche suitability of Mastomys natalensis (M. natalensis) and Lassa virus for the current period, 2030, 2050, and 2070. Each future projection (i.e., for 2030, 2050, and 2070) was performed according to four different bias-adjusted global climate models and three different representative concentration pathways (RCPs), i.e., greenhouse gas concentration scenarios considered by the Intergovernmental Panel on Climate Change (IPCC): RCP 2.6, RCP 6.0, and RCP 8.5. Here, we only report the projections obtained under RCP 6.0 (see Fig. S1 for the other scenarios as well as for the standard deviations associated with all projections, and see Fig. S2 for explicit differences between current and future

projections). For a specific time period, we report ecological niche suitability averaged over the projections obtained with the four different climatic models (see the text for further detail). **b** Projections of the human population at risk of exposure to Lassa virus for the current period, 2030, 2050, and 2070. For those estimations, we also re-estimate these projections while fixing the human population, i.e., not using the future projections of human population to estimate the number of people at risk (see also Fig. S3 for spatially-explicit estimation of future human exposure to Lassa virus and Fig. S6 for the estimations of the human population at risk of exposure to Lassa virus under all RCP scenarios). Source data are provided as a Source Data file.

predominantly in current endemic countries of Guinea, Sierra Leone, Liberia, and Nigeria (Fig. 2a).

At future time points, we found that the ecological niche of Lassa virus may substantially expand under both RCP 6.0 and RCP 8.5 (Fig. 2a, Fig. S1). RCP 2.6 and RCP 8.5 are the most extreme scenarios and refer to either stringent mitigation (RCP 2.6), or high-end emissions (RCP 8.5), while RCP 6.0 represents a medium-high emission scenario[59]. Focusing on RCP 6.0, we projected that by

2070, most of the region between Guinea and Nigeria will become suitable (ecological suitability >0.5) for Lassa virus (Fig. 2a, see Fig. S1 for the other scenarios as well as for the standard deviations associated with all projections). In addition, we found that several regions will likely become suitable in Central Africa, including in Cameroon and the Democratic Republic of the Congo (DRC), but also in East Africa, notably in Uganda. For *M. natalensis*, we found that irrespective of the scenario, the ecological niche will likely

remain stable in range, with suitability values that increase over time and across the entire niche (Fig. 2a and Fig. S1). These results show that, considering a medium-high scenario of evolution of global warming (RCP 6.0), the ecological niche of Lassa virus may expand well beyond current endemic countries, notably into parts of Central and East Africa.

To investigate the factor(s) driving the expansion of the niche of Lassa virus, we represented, on a map of Africa, the environmental values for the main factors influencing ecological suitability at current and future time points (Figs. S4, S5). In Central and East Africa, areas showing an increased suitability for the virus under RCPs 6.0 and 8.5 also exhibited an increase in temperature and pastures/rangeland land coverage (Fig. 2a, Figs. S4, S5). Based on our observations, these two factors may thus primarily drive the expansion of the range suitable for Lassa virus.

## A predicted increase in the population living in Lassa virus suitable areas

Our projections show that the potential expansion of the ecological niche of Lassa virus would occur in regions where *M. natalensis* is (and will remain) present, as indicated by our estimates of present and future suitability for *M. natalensis* across Africa (refs. 34,46–48 and Fig. 2). In the next decades, with *M. natalensis* being present where ecological conditions will be suitable for virus circulation outside of the current endemic range, the distribution of Lassa virus in Africa could potentially widen. The recent emergence of Ebola virus in West Africa and of West Nile virus in North America illustrate how zoonotic viruses can travel over long distances to effectively settle in new regions[50–54], dramatically widening their range, and putting more people at risk of infection.

We next investigated how the expansion of the ecological niche of Lassa virus could affect the future number of people at risk of infection. To estimate the current and future human population in the virus niche, we considered population projections in areas with an estimated ecological suitability above 0.5 (Fig. S3). We focused again on three future time points (2030, 2050, and 2070), and three climate scenarios (RCPs 2.6, 6.0, and 8.5). We found that under RCP 6.0, the human population living in the niche of Lassa virus, where conditions are suitable for virus circulation, may increase from 92 million today (95% highest posterior density [HPD] interval: [83–98]) to 453 [414–497] million by 2050, and to 700 [624–779] million by 2070 (Figs. 2b and S6, Table S1). This increase however, may be driven by demographic growth in current suitable areas rather than by the spatial expansion of the virus ecological niche[48]. To investigate this, we first examined current population numbers in areas suitable for Lassa virus in 2070 (scenario RCP 6.0) and found that they are currently home to 179 million people [159–199]. This result suggests that the population is expected to grow substantially throughout the entire niche of the virus (as projected in 2070), which will more than triple by 2070 (Fig. 2b, Table S1). When comparing the number of people that will live in current or future parts of the niche in 2070, we found that population growth should be comparable in both areas (Table S1). More specifically, our results show that by 2070, 363 [333–384] million people may be exposed to Lassa virus infection in current suitable areas and that expansion of the ecological niche of the virus might put 337 [260–405] million more people at risk of infection. As our estimates may be sensitive to the arbitrary cut-off value of 0.5 we used to define an area as ecologically suitable, we repeated our analysis considering different arbitrary cut-off values (0.25 and 0.75), and observed the same trends (Table S1).

## Lassa virus circulation is remarkably slow in endemic areas

In our ecological niche modelling analyses, we found that within a few decades, ecological conditions will be suitable for Lassa virus circulation beyond its current endemic range in West Africa. If Lassa virus is introduced into a new suitable region, we estimated that tens to hundreds of millions more people may be at risk of infection. The virus may however need time to spread locally and occupy a substantial area. To assess how fast the virus may spread following a potential future introduction into a suitable environment, we analysed the spatiotemporal spread of the virus using geotagged viral genomes. We showed that Lassa virus dispersal in endemic areas is remarkably slow compared to other zoonotic viruses.

To infer the spatiotemporal spread of Lassa virus since the emergence of the four major clades[32], we analysed publicly available genomic sequences associated with a sampling date and location using a spatially-explicit Bayesian phylogeographic approach[62]. The genome of Lassa virus is segmented into a large (L) and a small (S) segment that may reassort during coinfections in *M. natalensis*[63,64]. As reassortment may result in distinct evolutionary histories for the L and S segments, we analysed them in separate phylogeographic inferences ($n$ = 255 and 411, respectively). We also divided our analyses between four main clades (Fig. S7): the "MRU clade" groups the subclades circulating in the Mano River Union (MRU) and Mali (also called lineages IV and V); "NGA clade II", "III" and "VI" correspond to the main clades circulating in Nigeria (also called lineages II, III and VI, respectively)[65]. The trees inferred by our phylogeographic analyses capture the spatiotemporal spread of the virus (Fig. 3), with each branch representing dispersal between an estimated start and end location, and associated with an estimated duration.

To investigate the spatial spread of the main clades, we represented the trees inferred by our phylogeographic analyses on maps, separately for the MRU and the Nigerian clades. We observed that in Nigeria, the main clades are confined to distinct areas: clades II and III circulate south and north of the Niger and Benue rivers, respectively, while clade VI is limited to states in the south west (Osun, Ekiti, Ondo, Kwara, Fig. 3). We also noted that sequences from the MRU clade grouped in three main clusters circulating respectively in eastern Sierra Leone, Guinea and Mali (Fig. 3). The strong geographic structure we observed in our phylogenetic trees was consistent across the S and L segments (Figs. 3, S9, and S10) and aligns with previous studies[30,31,66]. These findings suggest that, although the spread of Lassa virus encompasses hundreds of years (Figs. 3, S9, and S10), virus diversity is distinct across different areas.

To approximate how fast Lassa virus circulates in endemic areas (Manor River Union and Nigeria), we estimated the weighted lineage dispersal velocity[67], which corresponds to the total distance covered by the dispersal events in our trees divided by the sum of their durations. We found that Lassa virus circulates with a weighted lineage dispersal velocity between 0.8 and 1.0 km/year (95% HPD interval for the S and L segments: [0.7-1.0] and [0.9-1.0]; Table 1). Consistently across the L and S segments, our estimates of the weighted lineage dispersal velocity for each of the main clades show that virus circulation is slowest for the MRU clade and fastest for the Nigerian clade II (Fig. S8). While we cannot exclude the hypothesis that one or several clades are associated with an increased transmissibility or shorter serial intervals accelerating virus dispersal, the generally low clade-specific estimates (<1.5 km/year) remain similar. These results highlight that Lassa virus circulation is slow, which may in part explain why the main clades are confined to different areas within overall suitable regions (Fig. 2).

To determine how slow the velocity of Lassa virus circulation was compared to other zoonotic viruses, we assembled and sorted all published estimates of weighted lineage dispersal velocities (Table 1). We found that Lassa virus exhibits the slowest lineage dispersal velocity after Nova virus, while Ebola virus appeared to be the fastest. Our results indicate that Lassa virus circulation in endemic areas is particularly slow compared to other zoonotic viruses, potentially due to the small scale of the movements of its reservoir[25,33].

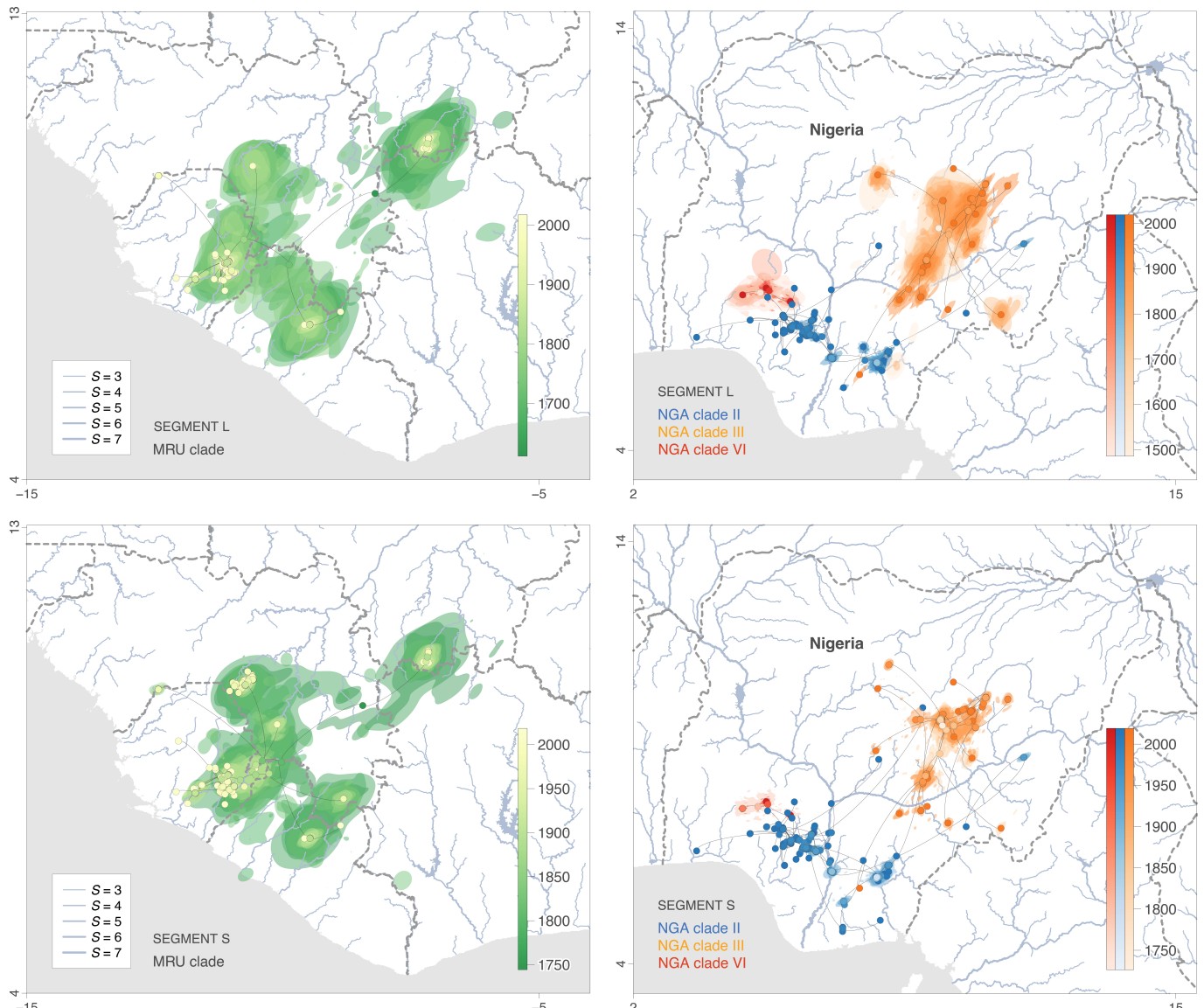

**Fig. 3 | Spatiotemporal diffusion of Lassa virus lineages in the western Africa region and Nigeria. Maximum clade credibility (MCC) tree obtained by continuous phylogeographic inference based on 1000 posterior trees.** A separate phylogeographic analysis was performed for segments L and S as well as, in the case of the Nigerian data set, on clades II, III, and VI. These MCC trees are superimposed on 80% highest posterior density (HPD) intervals reflecting phylogeographic uncertainty. Nodes of the trees, as well as HPD regions, are coloured according to their time of occurrence, and oldest nodes (and corresponding HPD regions) are here plotted on top of youngest nodes. The trees are superimposed on maps displaying the main rivers present in the study area and classified according to their Strahler number $S$, which measures the importance of a river by looking at the number of upstream rivers connected to it. International borders are represented by grey dashed lines. See also Figs. S9 and S10 for visualisations clade by clade. The "MRU clade" groups the subclades circulating in the Mano River Union (MRU) and Mali (also called lineages IV and V); "NGA clade II", "III" and "VI" correspond to the main clades circulating in Nigeria (also called lineages II, III and VI, respectively).

## Limited impact of environmental factors on Lassa virus dispersal dynamics

Our phylogeographic inferences show that Lassa virus circulation is remarkably slow in endemic areas, which may explain, at least in part, why the spatial spread of the main clades is limited, even within overall suitable regions. This finding suggests that, in case of introduction into a new suitable area, Lassa virus may require time to spread locally. In addition, since Lassa virus depends on *M. natalensis* for transmission, any environmental feature limiting the mobility of the reservoir may also impact virus dispersal. Main waterways in particular, have been proposed to act as barriers preventing the spread of Lassa virus, based on phylogenetic evidence that virus diversity is distinct across different sides of the Niger and Benue rivers in Nigeria[30,66] (Fig. 3). For other viruses, such as rabies, there is evidence that environmental factors including elevation or croplands coverage have an impact on virus lineage dispersal velocity[68,69].

To determine if main waterways act as barriers to virus dispersal, we investigated whether Lassa virus tended to avoid crossing rivers based on our phylogeographic reconstructions. Using a least-cost path algorithm[70], we computed the cost for the virus to travel through a landscape crossed by rivers based on both stream network data (Table S6) and the virus dispersal trajectory. We compared the cost of the observed spread inferred by our phylogeographic analyses to the cost computed under a null dispersal model that is unaware of rivers, and then estimated the statistical support for our test by approximating a Bayes Factor (BF) in favour of a cross-avoiding behaviour. We repeated our test for a range of stream sizes considering different threshold values of the Strahler number ($S$) − a proxy for river stream

**Table 1 | Comparison of lineage dispersal velocities estimated for different data sets**

| Data set | Weighted lineage dispersal velocity | Sampled sequences | Reference |
|---|---|---|---|
| Nova virus (moles), Belgium | 0.3 km/year [0.3, 0.4] | 100 | Laenen et al.[67] |
| Lassa virus, segment L, Africa | 0.8 km/year [0.7, 1.0] | 254 | (Present study) |
| Lassa virus, segment S, Africa | 1.0 km/year [0.9, 1.0] | 410 | (Present study) |
| Rabies virus (skunks), USA | 9.4 km/year [8.3, 10.6] | 241 | Kuzmina et al.[68] |
| Rabies virus (raccoons), USA | 11.8 km/year [9.6, 13.3] | 47 | Biek et al.[69] |
| Rabies virus (bats), eastern Brazil | 12.5 km/year [7.8, 20.3] | 41 | Vieira et al.[70] |
| Rabies virus (dogs), northern Africa | 16.8 km/year [14.0, 19.7] | 250 | Talbi et al.[71] |
| Rabies virus (bats), Peru | 17.7 km/year [14.6, 21.1] | 260 | Streicker et al.[72] |
| Rabies virus (mainly dogs), Iran | 18.1 km/year [16.3, 20.8] | 105 | Dellicour et al.[73] |
| Rabies virus (bats), Argentina | 34.7 km/year [28.1, 41.6] | 131 | Torres et al.[74] |
| H5N1 virus, Mekong region | 149.0 km/year [115.9, 170.2] | 320 | Dellicour et al.[75] |
| West Nile virus, North America | 165.0 km/year [158.0, 169.2] | 801 | Dellicour et al.[76] |
| Yellow fever virus, Brazil | 169.4 km/year [131.7, 214.4] | 99 | Hill et al.[77] |
| Porcine deltacoronavirus, China | 184.7 km/year [134.7, 234.4] | 97 | He et al.[78] |
| Ebola virus, West Africa | 598.1 km/year [556.4, 635.3] | 722 | Dellicour et al.[79] |

For each data set, we report both the posterior median estimate and the 95% highest posterior density (HPD) interval in kilometres per year (km/year).

size, based on a hierarchy of tributaries[71]. We found only moderate evidence (3 <BFs <20)[72] that the virus dispersal trajectory tends to avoid crossing rivers, no matter the Strahler number used as cut-off value for the stream size (Table S2). Overall, our results provide no strong evidence that waterways may act as notable barriers to the dispersal of Lassa virus.

We next examined how environmental conditions may affect the velocity of Lassa virus circulation considering a set of nine environmental factors for which we collected geo-referenced data from public databases (Fig. S11, Table S6). For all virus dispersal events inferred by our phylogeographic analyses, we investigated whether the duration of the dispersal correlated with the environmental factors in our testing set. To assess these correlations, we computed an "environmental distance", which corresponds to the distance of the dispersal event, weighted according to the environmental conditions along the path of dispersal. Our procedure only considers constant-in-time environmental values that do not reflect the climatic and land cover conditions during the earliest part of Lassa virus dispersal history, so we restricted our analyses to the most recent dispersal events (corresponding to tip branches of the trees from our phylogeographic reconstructions; Fig. S7). On average, this corresponds to considering lineage dispersal events starting in 1991 for segment L (95% HPD: [1988–1995]) and 1993 for segment S (95% HPD: [1991–1994]). We only found moderate evidence (3 <BFs <20) that the presence of savannas may slow down viral circulation (Tables S3 and S4). These results suggest that the environmental factors considered in our analysis have no dramatic impact on the velocity of Lassa virus circulation.

**Limited virus propagation following introduction into a new suitable area**

In the post hoc analyses of our phylogeographic inferences, we did not identify any environmental factor that may prevent or notably slow down virus spread in a suitable environment. Hence, in case of introduction into a new suitable region, the main parameter that we can expect to limit Lassa virus propagation based on our analysis would be its slow lineage dispersal velocity. To illustrate how a slow lineage dispersal velocity may limit the spatial extent of virus spread following a potential introduction, we simulated virus dispersal based on the parameters inferred by our phylogeographic analyses (Fig. 4). We ran simulations over a 20-year period in two areas: one projected to become suitable for virus circulation by 2050 under scenario RCP 6.0 and the other one, under RCP 8.5. To simulate virus dispersal, we randomly sampled dispersal events inferred by our phylogeographic

analyses for the Nigerian clade II, for which we have the largest number of sequences. We set the trajectory of dispersal events by selecting the ending location with a probability equal to the local ecological suitability (as projected in our ecological niche modelling analyses). By mapping the results of 1000 simulations of virus dispersal, we show that Lassa virus would likely remain confined within a range of ~200 km² (Fig. 4), even when starting within a large suitable area (e.g., with scenario RCP 8.5, Fig. 4 and Fig. S14). Our simulations show how, if Lassa virus circulation is as slow as in current endemic areas, virus propagation would remain spatially limited over the first decades following its introduction into a new ecologically suitable area.

## Discussion

Previous molecular dating studies have shown that Lassa virus has been circulating for at least 1000 years and originated in present-day Nigeria, from where it spread to the West, reaching into the MRU region[14,32,73,74]. Lassa virus is considered endemic in Guinea, Liberia, Nigeria, and Sierra Leone, but the virus likely circulates in other neighbouring countries along its presumed dispersal path[14–17]. Although we do not attempt to precisely map the current range of Lassa virus, our ecological suitability estimates for the current period appear globally similar to the results obtained in an earlier study[34] and show areas suitable for virus circulation across West Africa (see Fig. 2a). Consistent with reports of Lassa virus infections in humans and rodents outside of endemic hotspots, our results suggest that the virus may be present in most coastal West African countries and Mali, prompting strengthened Lassa fever surveillance throughout the whole region.

*M. natalensis* is considered the primary reservoir of Lassa virus[24] and it is still unclear why the distribution of this rodent species extends far beyond that of the virus, which is limited to West Africa[33]. Our analyses show that different environmental factors determine ecological suitability for the virus and its host, suggesting that the absence of Lassa virus beyond West Africa could be partly due to environmental constraints. To be able to estimate future ecological suitability for Lassa virus circulation, we have used projected environmental data with a low resolution (0.5 decimal degrees), due to the coarse scale of climate change projections[59,75]. This limited resolution reduced our ability to account for small-scale environmental variations that could affect suitability for Lassa virus; however, the good performance of our models (area under the receiver operating characteristic curves between 0.74 and 0.85) suggests that our approach provides a reasonable estimate of the distribution of Lassa virus infections.

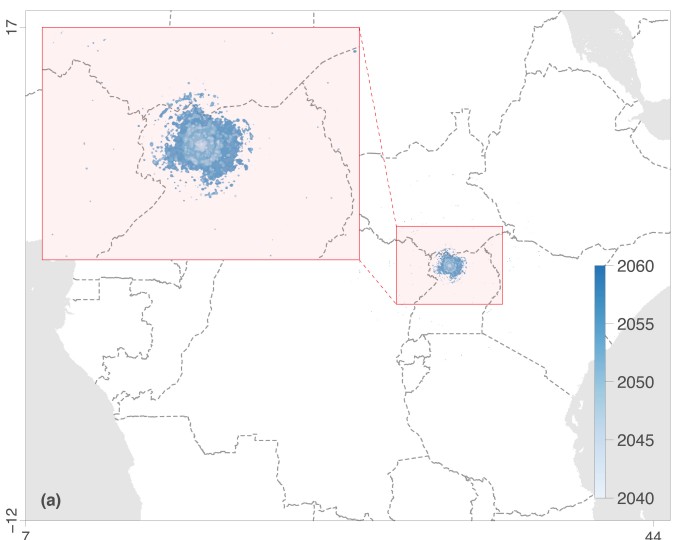
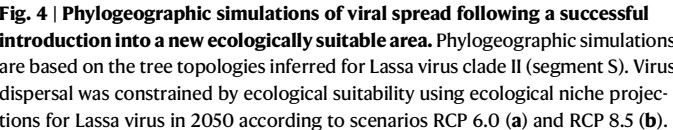
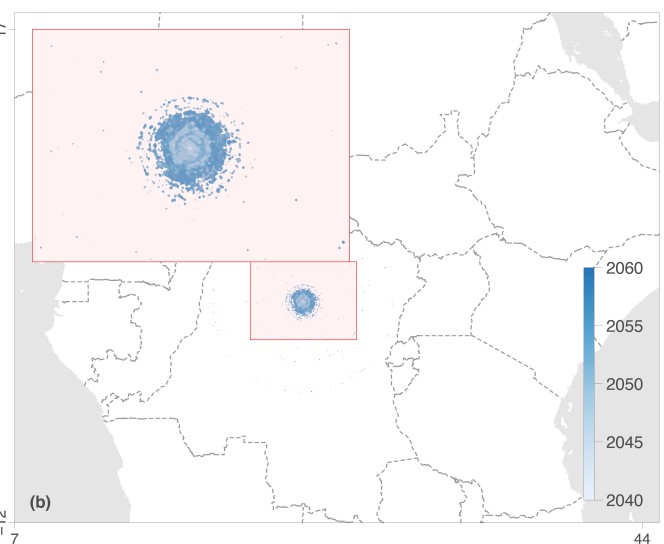

**Fig. 4 | Phylogeographic simulations of viral spread following a successful introduction into a new ecologically suitable area.** Phylogeographic simulations are based on the tree topologies inferred for Lassa virus clade II (segment S). Virus dispersal was constrained by ecological suitability using ecological niche projections for Lassa virus in 2050 according to scenarios RCP 6.0 (**a**) and RCP 8.5 (**b**).

95% highest posterior density (HPD) polygons are coloured according to time and based on 1000 simulations starting from the same ancestral location. For each set of simulations, a zoom on the outcome is shown. For the illustration, five distinct phylogeographic simulations per scenario are also displayed in Fig. S14.

In addition to environmental aspects, several other factors may also contribute to the difference in distribution between Lassa virus and its reservoir. As reported for other mammarenaviruses, the virus may only be present in the *M. natalensis* subtaxon A-I[36,37]. Records of Lassa virus infection in *M. natalensis* subtaxon A-II[76] and in other rodent species including *M. erythroleucus* or *Hylomyscus pamfi*[77] suggest, however, that susceptibility to Lassa virus infection may not be species or subtaxon specific. Other possible explanatory factors include intra-host competition between different viruses[38,78], or cross-immunity due to the circulation of closely related viruses[40,41]. As most other old world arenaviruses that circulate in *M. natalensis* are found only in East Africa[12,36,37], there is little data to assess these two mechanisms based on field data. Only in Mayo-Ranewo (Taraba state, eastern Nigeria), rodent trapping studies have identified a Mobala-like virus in *M. natalensis*[76,79], which does not seem to effectively restrict the transmission of Lassa virus, as human infections are often reported in that state of Nigeria[80].

In our phylogeographic analyses we find that Lassa virus mostly spreads on a small spatial scale, with relatively few long-distance dispersal events (Fig. 3) but we do not identify environmental factors that seem to strongly restrict or slow down virus spread. Using a phylogeographic simulation procedure, we also show that a slow lineage dispersal velocity would likely result in a limited spatial propagation if Lassa virus was successfully introduced in a new ecologically suitable area. The slow spread of Lassa virus may be due to the small scale of the movements of its reservoir, as suggested by genetic studies showing that *M. natalensis* rodents travel rarely outside of their commensal habitat and are prone to high levels of consanguinity[25,33]. However, it is surprising that Lassa virus —— and by implication its reservoir —— seem unrestricted by the environment. Of note, our results were not always consistent between the L and S segments, possibly due to the lower number of L sequences (255) in our data set compared to S sequences (411). More generally, the number of genomic sequences in our datasets may offer limited power for the tests we used to assess the possible impact of rivers and other environmental factors on virus spread. Hence, a larger sampling of Lassa virus genomes throughout the virus range would allow for better evaluation of the role of the environment in limiting spread. Another caveat here is that our procedure considers

constant-in-time environmental values that do not reflect changes in climatic and land cover conditions that may have occurred throughout the virus dispersal history. Indeed, it is estimated that in the past century alone, about 10 million ha of forest were lost in West Africa, mostly to the benefit of agricultural expansion[81]. Over the same period, annual precipitation has decreased throughout the area while temperatures have risen, up to over 1 °C in some parts of Nigeria[82]. To address this issue, we restricted our analysis to the most recent dispersal events in the virus dispersal history which date back, on average, to the 1990s. However, our approach still does not account for abrupt changes in climate, land use, and population density that may have occurred throughout the last decades in the West African region[56].

In our study, we use phylogeographic simulations to highlight how, in the absence of restrictions from the environment, a slow lineage dispersal velocity may limit the propagation of Lassa virus in case of a successful introduction into a new ecologically suitable area. We use these simulations for illustration and not prediction as the dispersal dynamics upon virus emergence in a new region are unclear. The virus may spread swiftly through an immunologically naïve rodent population, but the low mobility of the rodent reservoir could still limit the velocity of virus dispersal on a larger scale. As underlined above, a number of other elements may also come into play, such as the nature of host species or subtaxa as well as potential cross-immunity or competition due to the local co-circulation of closely related viruses. Nevertheless, it is worth pointing out that following the emergence of Lassa virus in the MRU, virus circulation remained as slow − if not slower − as in Nigeria, as highlighted by our estimates of weighted lineage dispersal velocity (Fig. S8).

Our ecological niche modelling analyses highlight a risk of expansion of Lassa virus towards regions in Central and East Africa that could potentially lead to a drastic increase in the number of people living in areas ecologically suitable for Lassa virus (Table S1). To reach the largest ecologically suitable regions we identify in DRC and Uganda, the virus would have to spread over several hundreds of kilometres and cross regions with low ecological suitability. Such long-distance movements likely allowed Lassa virus to reach the Mano River Union from Nigeria several hundred years ago[32]. This early part of the virus dispersal history, however, remains poorly understood and it is

thus hard to predict if the virus is likely to travel across the African continent again. In addition, some factors may hinder virus dispersal following a hypothetical introduction into a new ecologically suitable region. First, in our analyses, as there is evidence that Lassa virus infection may not be species or subtaxon-specific[76,77], we considered the entire *M. natalensis* species to be susceptible to Lassa virus infection. It is, however, possible that the virus spreads less (or more) efficiently in a different subtaxon of *M. natalensis*. Second, while in most of the areas where ecological suitability for LASV appears to increase over time outside of West Africa (e.g., DRC, Uganda, Cameroon), no *Mammarenavirus* species have so far been identified[12], we cannot rule out the possibility that potential cross-immunity or competition due to the local co-circulation of closely related viruses may hinder the propagation of LASV.

To provide a very conservative estimate of the future risk of exposure to Lassa virus, we can focus on population growth in the endemic range and leave aside its possible expansion. When considering a cut-off value of 0.5 for the ecological suitability threshold defining the niche of the virus, we estimated that population growth in endemic countries could alone put 186 (95% HPD interval: [172–196]) and 341 [315–369] million people at risk of infection by 2030 and 2070, respectively (Table S1), compared to an estimated 83 [76–87] million today. A limitation to these estimates is that our population projections do not take into account migrations due to environmental and climate change pressures, which could affect projections in regions where extreme weather conditions are expected.

In this analysis, we focused on evaluating the population that will likely be exposed to Lassa in endemic regions or that might be exposed in regions that we predict to be suitable for Lassa virus in the future. Importantly, we did not evaluate how much heightened surveillance, improved sanitation, and increased awareness may help to reduce exposure to Lassa virus in the future[83–86]. These measures will, however, remain crucial to prevent infections. Even more so as we report an extremely slow dispersal velocity of Lassa virus in endemic areas, suggesting that localised efforts that target infection hotspots may prove to be highly effective.

A large part of the population growth expected in endemic areas is driven by Nigeria (~91%), a country that has reported an unusual increase in the number of reported Lassa fever cases over the last two years[80,87]. This uptick was not attributed to increased inter-human transmission[30,31] or to the emergence of a specific viral strain[30,31]; but raised the question of a more intense circulation within the reservoir or of an improvement in surveillance and public awareness. To discriminate between these two hypotheses, we investigated the evolution of the overall genetic diversity of Lassa virus in the main Nigerian clade (clade II) over the past decades, using a coalescent approach that accounts for preferential sampling. We found that the effective population size of Nigeria clade II increased over the last years (segment S; Fig. S13), suggesting that the recent uptick in cases in Nigeria was not the sheer result of an improvement in surveillance. Hence, even if Lassa virus does not expand to new regions in the near future, the virus still actively circulates in increasingly populated endemic areas, and there is thus an urgent need for more efficient prophylactic and therapeutic countermeasures.

With anthropogenic climate change and an increasing impact of human activities on the environment, extensive studies of the ecology and spread of zoonotic and vector-borne diseases are needed to anticipate possible future changes in their distribution[88,89]. We showed that changes in temperature, precipitation, and pastures/rangeland land coverage may expand the ecological niche of Lassa virus beyond current endemic areas, potentially exposing hundreds of million more people to Lassa. By simulating virus spread, we highlight that, in the scenario of a successful introduction and propagation of the virus in a new ecologically suitable area, the emerging circulation foci could remain limited to a small spatial scale over the first decades. Our study provides an example of how ecological niche modelling and spatially-explicit phylogeography can be effectively combined to investigate the future risk of a major zoonotic disease.

## Methods

### Ecological niche modelling of Mastomys natalensis and Lassa virus

We employed the boosted regression trees[90] (BRT) approach implemented in the R package dismo[58] (version 1.3-8) to perform ecological niche modelling analyses of both Lassa virus and its host, *M. natalensis*. BRT is a machine learning method that allows to model complex non-linear relationships between the probability of occurrence and various predictor variables[58,91]. This approach aims to generate a collection of sequentially fitted regression trees that optimise the predictive probability of occurrence based on predictor values[90,91], which can also be interpreted as a measure of ecological suitability. In a comprehensive review of distribution modelling methods, Elith et al.[90] found BRT to perform best along with the maximum entropy method[92].

The BRT approach requires both presence and absence data. When unavailable, as this is the case for Lassa virus and its host, absence data can be approximated by random pseudo-absence points sampled from the study area (also referenced as the "background"). For Lassa virus, we only sampled pseudo-absences in raster cells in which the presence of *M. natalensis* has been recorded. While heterogeneous disease surveillance efforts likely bias the spatial distribution of Lassa records[46,47], this procedure avoids treating undersampled areas as ecologically unsuitable for the virus, but also limits the potential impact of such heterogeneity in sampling effort or surveillance[93–95]. Of note, while several rodent species such as *M. erythroleucus*[77] may be important for Lassa virus transmission, we chose to be conservative and focused on the main species recognised as a reservoir host for Lassa virus to define our background. Similarly, for *M. natalensis*, we only sampled pseudo-absences in raster cells in which the presence of at least one individual of another species belonging to the Muridae family has been recorded. Because it only requires a single occurrence record to consider a presence, we discarded all but one occurrence record per raster cell. We applied the same filtering step for the pseudo-absence points and simply discarded pseudo-absences falling in raster cells with occurrence data. This filtering procedure is required to only have a presence or a pseudo-absence assigned to each grid cell, but also has the advantage of minimising the risk of artefacts due to spatially heterogeneous disease/wildlife surveillance efforts in the resulting BRT models. Indeed, even in areas where the disease surveillance was limited, it only necessitates at least a single occurrence record to consider the presence of the target species in the large grid cell we use.

To select the optimal number of trees in the BRT models, we used a spatial cross-validation procedure based on five spatially separated folds generated with the blockCV R package[95] (version 2.1.4). We employed a spatial rather than a standard cross-validation because the latter may overestimate the ability of the model to make reliable predictions when occurrence data are spatially auto-correlated[97], which can frequently be the case. All BRT analyses were run and averaged over 10 cross-validated replicates, with a tree complexity set at 5, an initial number of trees set at 100, a learning rate of 0.005, and a step size of 10. We evaluated the inferences using the area under the receiver operating characteristic (ROC) curve, also simply referred to as "area under the curve" (AUC). Among replicates, AUC values ranged from 0.68 to 0.73 for *M. natalensis* (mean = 0.71), and from 0.74 to 0.85 for Lassa virus (mean = 0.79).

We obtained occurrence data for *M. natalensis* species from the Global Biodiversity Information Facility (GBIF.org, https://doi.org/10.15468/dl.hrjyj1, accessed 2019-07-19), the Integrated Digitized Biocollections (https://www.idigbio.org, accessed 2020-01-04), the

Field Museum of Natural History Zoological collections (https://collections-zoology.fieldmuseum.org, accessed 13 December 2019), and the African Mammalia database (http://projects.biodiversity.be/africanmammalia, accessed 2019-12-14). This data set was supplemented with the data available in the scientific literature (search for term "*Mastomys natalensis*", in PubMed and Google). Duplicate records as well as records located in the ocean were excluded from the final data set, totalling 2504 unique *M. natalensis* occurrence records. For 26 of those records, the location was not provided as spatial coordinates but as a locality (below or at the administrative level 4). Therefore, the latitude and longitude data correspond to that of the locality (determined as described in the subsection *Selection and preparation of viral sequences*; see below). Occurrence data for the *Muridae* family were obtained from the GBIF database (GBIF.org, https://doi.org/10.15468/dl.cs3c41, accessed 2019-07-19). Duplicate records and records located in the ocean were excluded from the data set, totalling 10,806 unique *Muridae* occurrence records for the African continent. Occurrence data for Lassa virus were obtained by combining the data set from Fichet-Calvet & Rogers[48] with records associated with sequences from our Lassa virus sequence data set (see below the subsection *Selection and preparation of viral sequences* for further detail), records of infected *M. natalensis* from our host occurrence data set and the data available in the scientific literature (search for term "Lassa virus" in PubMed). Duplicate records were discarded from the data set, resulting in 310 unique Lassa virus occurrence records. For two of those records, the location was not provided as spatial coordinates but as a locality (below or at the administrative level 4) so the latitude and longitude data corresponded to that of the locality (determined as described in the subsection *Selection and preparation of viral sequences*; see below). Our BRT models were trained on current environmental factors and then used to obtain estimates of future ecological niches for both Lassa virus and *M. natalensis*.

The BRT analyses were based on several environmental factors: harmonised present-day and future climate, land cover, and population data available through the Inter-Sectoral Impact Model Intercomparison Project phase 2b (ISIMIP2b)[59]. The climate information consists of daily gridded near-surface air temperature and surface precipitation fields derived from four bias-adjusted[98] global climate models (GCMs; GFDL-ESM2M[99], HadGEM2-ES[100], IPSL-CM5A-LR[101], and MIROC5[102]) participating in the fifth phase of the Coupled Model Intercomparison Project (CMIP5[103]). We considered simulations conducted under historical climate forcings and RCPs 2.6, 6.0, and 8.5. In addition, we considered observed gridded temperature and precipitation from the concatenated products GSWP3 and EWEMBI[58] for assessing the current (1986-2005) conditions. For land cover, we used version 2 of the Land Use Harmonisation (LUH2[104]) providing historical and projected land cover states under a range of shared socio-economic pathways (SSPs), and from which we considered SSP1-26, SSP4-6.0, and SSP8-85. Finally, we retrieved gridded population projections[105] under SSP2-26. For each combination of product (GCM, GSWP3-EWEMBI LUH2, gridded population), scenario (historical, RCP, SSP), and analysis window (1986-2005, 2021-2040, 2041-2060, and 2061-2080), we computed the grid-scale temporal mean. For each scenario and time period, we estimated an index of human exposure (IHE) which corresponds to human population estimates (log$_{10}$-transformed) in raster cells associated with an ecological suitability for Lassa virus above or equal to 0.5. Specifically, we used these IHE values to calculate the number of people at risk of exposure to Lassa virus. To investigate the specific effect of human population growth in current and future suitable areas, we also re-estimated future IHE values using (i) current population estimates with future projections of ecological suitability for Lassa virus to estimate population growth throughout current and future suitable areas, and (ii) future projections of human population with current projection of ecological suitability for Lassa

virus to estimate the future population living in current suitable areas (Table S1). For each estimate, we calculated the mean and 95% HPD interval across all climatic models and ecological niche model replicates.

## Selection and curation of viral sequences
All publicly available sequences for Lassa virus were downloaded from the NCBI Nucleotide database (keywords: "lassa NOT mopeia NOT natalensis"; $n = 729$ L and 1202 S sequences; database accessed on October 31, 2019). They were combined with recently generated sequences from Nigeria that have been sequenced using the MinION technology (Oxford Nanopore) in conjunction with a non-targeted metagenomic RNA sequencing approach[31] and that are publicly available on the website virological.org (https://virological.org/t/2019-lassa-virus-sequencing-in-nigeria-final-field-report-75-samples/291).
We filtered the data by: (i) excluding laboratory strains (adapted, passaged multiple times, recombinant, obtained from antiviral or vaccine experiments), (ii) excluding sequences without a timestamp, (iii) keeping only sequences from a single timepoint (if multiple timepoints were available for a patient), (iv) removing duplicates (when more than one sequence was available for a single strain), and (v) excluding sequences from identified hospital epidemics or sequences for which the location corresponded to the site of hospitalisation. The remaining sequences were trimmed to their coding regions and arranged in sense orientation separately for the S segment (NP-NNN-GPC) and the L segment (L-NNN-Z). The sequences were aligned using MAFFT[106] (version 7) and inspected manually using the program AliView[107] (version 1.26). At this step, we discarded low-quality sequences (manual curation) and very short sequences (combined ORF length <500nt). Since there is an overlap between the sequence data from the work of Siddle and colleagues[30] and of Kafetzopoulou and colleagues[31], we excluded sequences with zero or one mismatch between the two sets of sequences to ensure that there would not be duplicates in our data sets. Two types of alignments were generated. The alignments with all curated sequences regardless of the availability of detailed location information included 756 S segment sequences and 551 L segment sequences, respectively. The alignments with detailed location information included 411 S segment sequences and 255 L segment sequences, respectively. For the sequences with detailed location information, when no spatial coordinates were provided but only a name, spatial coordinates were determined using a combination of online platforms (Table S5). When several coordinates were available for one location, those matching across several data sets were kept, if the location was found in only one data set, the coordinates corresponding to the highest administrative level were kept.

## Inferring the dispersal history of Lassa virus lineages
We performed spatially-explicit phylogeographic reconstructions using the relaxed random walk (RRW) diffusion model[62] implemented in BEAST 1.10[96], which was coupled with the BEAGLE 3 library[108] to improve computational performance. We modelled the nucleotide substitution process according to a GTR + Γ parameterisation[109] and branch-specific evolutionary rates according to a relaxed molecular clock with an underlying log-normal distribution. These phylogeographic analyses were based on the alignments of sequences associated with known spatial coordinates. For both the demographic and phylogeographic reconstructions, we ran a distinct BEAST analysis for each segment (L and S), sampling Markov chain Monte-Carlo (MCMC) chains every 10$^5$ generations. We used Tracer 1.7[110] for identifying the number of sampled trees to discard as burn-in, but also for inspecting the convergence and mixing, ensuring that estimated sampling size (ESS) values associated with estimated parameters were all >200. We used TreeAnnotator 1.10[96] to obtain a maximum clade credibility (MCC) tree for each BEAST analysis. Finally, we

used the R package seraphim[67] to extract the spatiotemporal information embedded within trees obtained by spatially-explicit phylogeographic inference, as well as to estimate the weighted lineage dispersal velocity.

## Impact of environmental factors on the dispersal dynamics of Lassa virus lineages

Based on the spatially-explicit phylogeographic reconstructions, we performed two different kinds of analyses to investigate the impact of several environmental factors on the dispersal history and dynamics of Lassa virus lineages. First, we tested the impact of main rivers acting as potential barriers to Lassa virus dispersal (see Table S6 for the source of the original rivers shapefile). For this purpose, we used the least-cost path algorithm[70] to compute the total cost for viral lineages to travel through a landscape crossed by rivers. This algorithm uses an underlying environmental raster to compute the minimum cost to move from one position to another. Here, we generated rasters by assigning a value of 1 to raster cells that were not crossed by a main river and a value of $1 + k$ to raster cells crossed by a main river (raster resolution: ~0.5 arcmin). Because the raster cells that were not crossed by a main river were assigned a uniform value of 1, $k$ thus defines the additional resistance to movement when the cell does contain such a potential landscape barrier[111,112]. In order to assess the impact of that rescaling parameter, we tested three different values for $k$: 10, 100, and 1000. Furthermore, as the notion of main river is arbitrary, we used different threshold values of the Strahler number $S$ associated with each river to select the main rivers to consider in each analysis. In hydrology, $S$ can be used as a proxy for stream size by measuring the branching complexity, i.e., the position of a river within the hierarchical river network. In practice, we compared the total cost computed for posterior trees with the total cost computed on the same trees along which we simulated a stochastic diffusion process under a null dispersal model[113]. Hereafter referred to as "simulated trees", these trees were obtained by simulating a relaxed random walk process along the branches of trees sampled from the posterior distribution obtained by spatially-explicit phylogeographic inference[113]. Because this stochastic diffusion process did not take the position of rivers into account, we can expect the total cost to be lower for inferred trees under the assumption that viral lineages did tend to avoid crossing rivers. For each inferred or simulated tree, we computed the total cost TC, i.e., the sum of the least-cost values computed for each phylogenetic branch considered separately. Each inferred TC value (TC$_{inferred}$) was then compared to its corresponding simulated value (TC$_{simulated}$) by approximating a Bayes factor (BF) support as follows: BF = [$p_e$/(1-$p_e$)]/ [0.5/(1-0.5)], where $p_e$ is the posterior probability that TC$_{simulated}$ > TC$_{inferred}$, i.e., the frequency at which TC$_{simulated}$ > TC$_{inferred}$ in the samples from the posterior distribution. The prior odds is 1 because we can assume an equal prior expectation for TC$_{inferred}$ and TC$_{simulated}$.

Next, we tested the impact of several environmental variables, again described as rasters, on the dispersal velocity of Lassa virus lineages (Table S6, Fig. S11): main rivers (as defined by selecting rivers with a $S$ value higher than 2, 3, 4, 5, and 6), forest areas, grasslands, savannas, croplands, annual mean temperature, annual precipitation, and human population density. Except for the generated river rasters (see above), all these rasters had a resolution of ~2.5 arcmin. For one-dimensional landscape features such as rivers, we had to resort to higher resolution rasters (~0.5 arcmin) to obtain sufficiently precise pixelations (rasterizations) for this non-continuous environmental factor. Raster cells assigned to rivers would otherwise be exceptionally large given the size of the study area, which could potentially lead to artefactual results. Environmental rasters were tested as potential conductance factors (i.e., facilitating movement) as well as potential resistance factors (i.e., impeding movement). For each environmental variable, we also generated several distinct rasters with the following formula: $v_t = 1 + k*(v_o/v_{max})$, where $v_t$ is the transformed cell value, $v_o$ the

original cell value, and $v_{max}$ the maximum cell value recorded in the raster. The rescaling parameter $k$ here allows the definition and testing of different strengths of raster cell conductance or resistance, relative to the conductance/resistance of a cell with a minimum value set to 1[114]. For each of the three environmental factors, we again tested three different values for $k$ (i.e., 10, 100, and 1000). The following procedure can be summarised in three successive steps[115]: (i) based on environmental rasters, we computed environmental distance for each branch in inferred and simulated trees. These distances were computed using two different algorithms: the least-cost path and Circuitscape algorithm, the latter using circuit theory to accommodate uncertainty in the route taken[116]. For computational tractability, high resolution river rasters were only tested with the least-cost path algorithm. (ii) We estimated the correlation between time durations and environmental distances associated with each phylogenetic branch. Specifically, we estimated the statistic $Q$ defined as the difference between the coefficient of determination obtained when branch durations are regressed against environmental distances computed on the environmental raster, and the coefficient of determination obtained when branch durations are regressed against environmental distances computed on a uniform "null" raster, i.e., a uniform raster with a value of "1" assigned to all its cells. We estimated $Q$ for each tree and we thus obtained two distributions of $Q$ values: one for inferred and one for simulated trees. We only considered an environmental raster as potentially explanatory if both its distribution of regression coefficients and its associated distribution of $Q$ values were positive[117]. (iii) We evaluated the statistical support associated with a positive $Q$ distribution (i.e., with at least 90% of positive values) by comparing it with its corresponding null distribution of $Q$ values based on simulated trees. We formalised this comparison by approximating a BF support as defined above, but this time defining $p_e$ as the posterior probability that $Q_{estimated} > Q_{simulated}$, i.e., the frequency at which $Q_{estimated} > Q_{simulated}$ in the samples from the posterior distribution[68]. For computational reasons, the main rivers rasters, which had to be associated with higher resolution (see above), were only tested as resistance factors with the least-cost-path algorithm.

## Phylogeographic simulations

We implemented a phylogeographic approach to simulate virus dispersal over a 20-year period following a successful introduction event within a new ecologically suitable area in 2050 under scenarios RCP 6.0 and RCP 8.5. We simulated viral lineage dispersal events by randomly sampling from the dispersal events inferred by our phylogeographic analyses. These simulations were performed under the assumption of no notable impact of underlying environmental factors. To set the trajectory of lineage dispersal events, we selected the ending location with a probability defined by the local ecological suitability. The starting point of those simulations was selected arbitrarily within the most suitable area of the extended part of the ecological niche estimated for Lassa virus in 2050, and was thus different for simulations performed under scenarios RCP 6.0 and RCP 8.5.

## Inferring the demographic history of Lassa virus lineages

We performed demographic reconstructions using the flexible skygrid coalescent model[118] implemented in BEAST 1.10[96]. The skygrid model allows to estimate the past evolution of the viral population effective size through time. For these analyses, we also modelled the nucleotide substitution process according to a GTR + Γ parameterisation[109] and branch-specific evolutionary rates according to a relaxed molecular clock with an underlying log-normal distribution[110]. In the case of NGA clade II for which we inferred a recent increase in the global effective population size, we also performed a preferential sampling analysis[119]. By modelling the sampling times as a process dependent on effective population size, this complementary analysis allows to explicitly take into account heterogeneous sampling density through time, which can improve estimates of global effective population size[119].

**Reporting summary**

Further information on research design is available in the Nature Research Reporting Summary linked to this article.

## Data availability

All source data used in this study are available at https://github.com/sdellicour/lassa_spreads (https://doi.org/10.5281/zenodo.6998624). The sources of the different raster files used in this study are provided in Table S6. We obtained occurrence data for the Muridae family from the Global Biodiversity Information Facility (http://www.gbif.org, accessed 2019-07-19, GBIF occurrence downloads https://doi.org/10.15468/dl.cs3c41). For the *M. natalensis* species, we obtained occurrence data from the Global Biodiversity Information Facility (http://www.gbif.org, accessed 2019-07-19, GBIF occurrence downloads https://doi.org/10.15468/dl.hrjyj1), the Integrated Digitized Biocollections (http://www.idigbio.org/portal (2020), Query: {"filtered": {"filter": {"and": [{"exists": {"field": "geopoint"}}, {"term": {"scientificname": "mastomys natalensis"}}]}}}, 4348 records, accessed on 2020-01-04T05:40:40.066945, contributed by 19 Recordsets), the Field Museum of Natural History Zoological collections (Field Museum of Natural History (Zoology) Mammal Collection https://collections-zoology.fieldmuseum.org/list?f%5B0%5D=ss_CatCatalog%3A%22Mammals%22&_ga=2.123662347.1070684726.1508778418-143671043.1493067972, accessed 2019-12-13), and the African Mammalia database (African Mammalia, http://projects.biodiversity.be/africanmammalia/search, accessed 2019-12-14). Full citations are provided in "Citations_rodent_occurrence_data.txt" available on the GitHub repository referenced above. This data set was supplemented with the data available in the scientific literature (search for term "Mastomys natalensis", in PubMed and Google). For each record used in this analysis, the specific record or collection ID is specified in the file "Mastomys_natalensis_RK050820.csv" or "Muridae_family_allData_RK220819.csv", both available on the GitHub repository referenced above. The sources used to retrieve sampling coordinates for Lassa virus samples are listed in Table S5. For Lassa virus, occurrence data were obtained from the scientific literature (search for term "Lassa virus", in PubMed and Google) and the source of each record used in this analysis is specified in the file "Lassa_virus_cases_RK070820.csv" available on the GitHub repository referenced above. The sources of the different raster files used in this study are provided in Table S6. Data for the environmental factors used in the BRT analyses was obtained from the Inter-Sectoral Impact Model Intercomparison Project phase 2b (ISIMIP2b, https://data.isimip.org/). LASV sequences analysed in the present study were available on GenBank before November 20, 2019, except for the LASV sequences from cases sampled during the year 2019 in Nigeria, which are are publicly available on the website virological.org (https://virological.org/t/2019-lassa-virus-sequencing-in-nigeria-final-field-report-75-samples/291). Accession numbers of selected genomic sequences are listed in the file "LASV_all_the_metadata.csv" available on the GitHub repository referenced above. All processed data (BRT models, BRT predictions, phylogeographic inferences, dispersal statistics estimations, and seraphim analyses) generated in this study are also available on the GitHub repository referenced above. Source data are provided with this paper.

## Code availability

All R scripts and related files needed to run all the ecological niche modelling and landscape phylogeographic analyses, as well as BEAST XML files, are publicly available at https://github.com/sdellicour/lassa_spreads (https://doi.org/10.5281/zenodo.6998624).

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

## Acknowledgements

The authors thank David Pigott for sharing their environmental suitability predictions for *Mastomys natalensis*. The work was supported by the European Research Council under the European Union's Horizon 2020 research and innovation programme under grant agreement no. 725422 — ReservoirDOCS (M.A.S., P.L.), the European Union's Horizon 2020 project MOOD under grant agreement no. 874850 (P.L., S.Dellicour), the Wellcome Trust through project 206298/Z/17/Z—The Artic Network (M.A.S., P.L.), the National Institute of Allergy and Infectious Diseases of the National Institutes of Health under award numbers U01AI151812 (K.G.A.), R01AI153044 (P.L. and M.A.S.), and U19AI135995 (K.G.A). S.Günther acknowledges support by the German Research Foundation (DFG, grants GU883/5-1 and GU883/5-2), and by the German Federal Ministry of Health through the WHO Collaborating Centre for Arboviruses and Hemorrhagic Fever Viruses at the Bernhard-Nocht-Institute for Tropical Medicine (agreement ZMV I1-2517WHO005) and the Global Health Protection Program (GHPP, agreements ZMV I1-2517GHP-704 and ZMVI1-2519GHP704). R.F.G. and D.S.G. acknowledge support from the Coalition for Epidemic Preparedness Innovation, the Wellcome Trust Foundation, and the European and Developing Countries Clinical Trials Partnership Programme. R.F.G. acknowledges support from the NIH (Grants R01AI132223, R01AI132244, U19AI142790, U54CA260581, U54HG007480, and OT2HL158260) and Gilead Sciences. B.V. was supported by a postdoctoral grant (12U7121N) of the Research Foundation—Flanders (*Fonds voor Wetenschappelijk Onderzoek*). S.Gryseels was supported by FED-tWIN OMEgA. P.L. acknowledges support from the Research Foundation — Flanders (*Fonds voor Wetenschappelijk Onderzoek — Vlaanderen*; grant no. G066215N, G0D5117N, and G0B9317N). S.Dellicour acknowledges support from the *Fonds National de la Recherche Scientifique* (F.R.S.-FNRS, Belgium; grant no. F.4515.22) and from the Research Foundation — Flanders (*Fonds voor Wetenschappelijk Onderzoek — Vlaanderen*; grant no. G098321N). The content is solely the responsibility of the authors and does not necessarily represent the official views of the National Institutes of Health. Computational resources have been provided by the *Consortium des Équipements de Calcul Intensif* (CÉCI), funded by the *Fonds de la Recherche Scientifique de Belgique* (F.R.S.-FNRS) under Grant no. 2.5020.11 and by the Walloon Region.

## Author contributions

Study design: R.K., W.T., G.D., M.A.S., P.L., K.G.A., and S.Dellicour. Sequencing: L.E.K., S.Duraffour, and S.Günther. Data collection and curation: R.K., L.E.K., W.T., S.Gryseels, A.K., K.G., M.Momoh, J.D.S., A.G., F.A, D.S.G., S.O., E.O.E., R.F.G., A.R.S., M.G.P, M.McGraw, L.D.H, and S.Dellicour. Data analysis: R.K., W.T., G.D., K.G., M.Z., B.V., and S.Dellicour. Writing - original draft preparation: R.K., W.T., P.L., K.G.A., and S.Dellicour. Writing - review and editing: R.K., W.T., B.V., K.G., M.Z., M.A.S., P.L., K.G.A., and S.Dellicour. Supervision: P.L., K.G.A., and S.Dellicour.

## Competing interests

R.F.G. is a co-founder of Zalgen Labs, a biotechnology company developing countermeasures for emerging viruses. The other authors declare no competing interest.

## Additional information

¹Department of Immunology and Microbiology, The Scripps Research Institute, La Jolla, CA 92037, USA. ²Department of Microbiology, Immunology and Transplantation, Rega Institute, Laboratory for Clinical and Epidemiological Virology, KU Leuven - University of Leuven, Leuven, Belgium. ³Bernhard Nocht Institute for Tropical Medicine, Hamburg, Germany. ⁴Department of Hydrology and Hydraulic Engineering, Vrije Universiteit Brussel, Brussels, Belgium. ⁵Institute of Biotechnology, Life Sciences Center, Vilnius University, Vilnius, Lithuania. ⁶Evolutionary Ecology group, Department of Biology, University of Antwerp, 2610 Antwerp, Belgium. ⁷Vertebrate group, Directorate Taxonomy and Phylogeny, Royal Belgian Institute of Natural Sciences, 1000 Brussels, Belgium. ⁸Eastern Technical University of Sierra Leone, Kenema, Sierra Leone. ⁹Viral Hemorrhagic Fever Program, Kenema Government Hospital, Ministry of Health and Sanitation, Kenema, Sierra Leone. ¹⁰College of Medicine and Allied Health Sciences, University of Sierra Leone, Kenema, Sierra Leone.

[11]Irrua Specialist Teaching Hospital, Irrua, Nigeria. [12]Faculty of Clinical Sciences, College of Medicine, Ambrose Alli University, Ekpoma, Nigeria. [13]Department of Microbiology and Immunology, Tulane University, School of Medicine, New Orleans, LA 70112, USA. [14]Zalgen Labs, LCC, Frederick, MD 21703, USA. [15]Global Virus Network (GVN), Baltimore, MD 21201, USA. [16]Department of Integrative, Structural and Computational Biology, The Scripps Research Institute, La Jolla, CA 92037, USA. [17]German Center for Infection Research (DZIF), Partner site Hamburg–Lübeck–Borstel–Riems, Hamburg, Germany. [18]Department of Bio-mathematics, David Geffen School of Medicine, University of California, Los Angeles, CA, USA. [19]Department of Biostatistics, Fielding School of Public Health, University of California, Los Angeles, CA, USA. [20]Department of Human Genetics, David Geffen School of Medicine, University of California, Los Angeles, CA, USA. [21]Scripps Research Translational Institute, La Jolla, CA 92037, USA. [22]Spatial Epidemiology Lab (SpELL), Université Libre de Bruxelles, CP160/12 50, av. FD Roosevelt, 1050 Bruxelles, Belgium. ✉e-mail: rklitting@scripps.edu; simon.dellicour@ulb.be

