## [Peer Review File · Nature Communications]

Predicting the evolution of the Lassa virus endemic area and population at risk over the next decadesREVIEWER COMMENTS

Reviewer #1 (Remarks to the Author):

This good and interesting paper about using niche modelling and viral phylogeography techniques to find the underlying driving risk factors for Lassa fever virus transmission in Africa, and then use forward projections of those drivers to estimate the changes to the endemic range of the virus. The paper is well written and uses recently developed techniques, which have also been used with other viral systems. It is also excellent that you have made your scripts and data (trees) available on Github.

I have only a few minor comments:

The main methods used were (a) niche models with boosted regression trees using reported occurrences of the virus and environmental factor data, and (b) phylogeographic models with viral sequence data and environmental factor data. Both of these methods seem very suitable and have been used in similar studies before, either by the authors or others, but for different viruses (the references in the manuscript are sufficient)

Niche models - as with most (all?) such studies disease occurrence data is used, and obviously if occurrences are under-reported then this will bias the results. Since the authors only have presence data, and not absence data they performed a suitable (and usual) background randomisation procedure (psuedo-absences) and only used spatial grid cells where there the main mouse host population was known.

Phylogeographic analyses - these were performed by first generating the time-scaled phylogenetic trees with spatial information then analysing the tree branch lengths and spatial diffusion paths with environmental rasters. The authors found that there were not any strong associations with the environmental predictors. This is only partly surprising, and as the authors explain could be due to the mobility of the main host species and generally slow dispersion rates of the virus.

What would be interesting to know though is - have those predictors changed through time from the temporal origin point (TMRCA) of the trees to the present ? Of course there might not be enough sampled sequences, temporal and spatial resolution to see whether this is an important effect or not, but it potentially might help explain the lack of signal (e.g. too averaged out in time or over long branch lengths). Therefore perhaps the authors could mention whether the predictors have substantially changed from say the 1900's or 1950's up until now in the discussion, and also mention (again) whether the predictors are expected to change at a higher or increasing rate from now into the future.

Reviewer #2 (Remarks to the Author):

This paper represents and analysis of the Lassa Fever and its host *Mastomys natalensis* in two parts. First from a 'niche modelling' approach where the authors examine the respective environmental niches of disease cases and known locations of the host. The second part is an analysis of sequence data where they examine the speed of spread of different variants of LASV.

There is clearly a lot of careful and detailed work, backed by considerable expertise, that has gone into this paper. First, I note that a quick search of Lassa Fever quickly brings one to a very similar analysis by Redding et al. 2016 (<https://besjournals.onlinelibrary.wiley.com/doi/full/10.1111/2041-210X.12549>) that uses again uses niche-based methods to capture the disease case and host niches and projects them to the year 2070. This along with various papers by Moses et al. and a

review by Gibb et al. (<https://pubmed.ncbi.nlm.nih.gov/28875769/>) need to be addressed to show how this paper is a progression from them. This goes beyond the request to cite some preferred papers; it is a small field so very similar papers and those more informational ones need to be cited.

Overall, the structure of the paper is quite loose and seems to be a list of a wide variety of analyses rather than a tightly defined 'story' of the main findings and the sensitivity tests the authors have done to understand and back up these results. The introduction is somewhat vague and quite limited and does not seem to introduce the reader to all aspects of the varied analysis that are subsequently used, leaving readers from specific disciplines in the dark about what is being done and why.

This work seems very much to be two separate manuscripts to me, loosely brought together, diverging around line 206. I would suggest nailing down the questions and constructing two careful and concise papers to address them.

One could be: How does the range of *Mastomys natalensis* alter over the next 50 years, given the variation we see in SSP/RCP scenarios? Without properly accounting for reporting biases in case data, any attempt to infer the 'niche' of the disease I believe is strongly flawed. After all what is the process causing there to be different disease case niche to the host niche? The authors mention it is not clear, so surely reporting biased is probably the most appropriate null expectation.

Furthermore, any expansion of *Mastomys natalensis* range needs to also consider host dispersal speed and mechanisms (especially if you are going to do this for the virus) and the authors have demonstrated they have great expertise in e.g. diffusion models so this surely will be relatively straight-forward. I think any proposed host range expansion would also need to consider edge effects in movement corridors but again, this should be fairly straight-forward. This work would need to build on the Redding et al. analysis, which doesn't seem too hard as there is much scope to expand here, to make a strong stand-alone paper.

The second main and very interesting finding is centred on the analysis of the sequence data, which as a separate analysis is both interesting and definitely adds to the current state of knowledge. The slow movement of Lassa relative to the small number of other data points the authors refer to is fascinating, and I think it can start to shed light on the limited geographic range of Lassa Fever in humans. I think a tightly defined and informative introduction, setting some clear questions and importantly introducing the reader to ideas that are examined e.g. phylogeography, how and why viruses spread etc. would be important here. I am not an expert on viral evolution, but having worked in organismal evolutionary biology I wonder how the relative fitness advantages of different virus strains impact these processes? If any mutations present have only a limited advantage i.e. the system is some kind of evolutionary equilibrium then the expectation would be of slow movement. In my naivety, some kind of key innovation would alternatively see a temporary faster movement will this mutation fix in the population, before re-establishing some kind of (albeit brief) equilibrium. So isn't the state of the system important when measuring these velocities? If this is not the case it would be interesting to know why.

The lack of environmental signal perhaps could be to do with the way that *Mastomys natalensis* is frequently moved around in agricultural produce, probably more in recent history due to increasing roads and mechanised transport. I am not sure if there is a way to uncover whether there is more of a signal for older branches than more recent, i.e. there is temporal degradation over time, just a thought.

Some specific points:

Line 47: Need to expand and at least mention vaccine development and status.

Line 50: Also very brief; needs to mention huge impact of historical under-reporting and biases in the data.

Line 57: Where is the reference for this statement, hugely unknown mechanism of spill-over, needs altering or strongly qualifying?

Line 70-74: Need to go into more details about the climate links e.g. referencing the large body of work by Fitchet-Calvet and colleagues.

Line 90: why are we going back into introduction after a heading saying 'Results'?

Line 98: The data are so biased here it would be stretch to say you are able to infer the actual disease niche. The difference here between the host and disease niche, is likely to be strongly influenced by the signal of disease surveillance. To account for this, the authors need to use a modelling approach such as hierarchical mixed models that you can better use to understand the system, not mainly for making predictions as is usually the case with BRTs. To draw any conclusions, one would need to think causally about the process that driven the patterns of cases seen and attempt to control for those unmeasured confounding factors, such as health-seeking behavior and availability of diagnostic tests. For instance, Nigeria has the highest capacity and is also the area where most of the data come from.

Line 112: I am not sure these are the best methods for this purpose. Also, this all seems like methods rather than results.

Line 191: Why this number – what influence do other numbers have?

Lines 193: This seems very low, again what is role of underreporting here?

Lines 200-204: I am very dubious about these numbers, it all stems from believing the LASV niche model and whether the host (or hosts) would actually spread, followed by the virus. Species tend to move relatively slowly with climate change (<https://www.science.org/doi/10.1126/science.1206432>) so how far would *Mastomys natalensis* have spread in this time? How does the boom-bust population dynamics of *Mastomys natalensis* look in these new environments, and how important are these population processes.

Lines 275-279: "major waterways may have limited impact on virus dispersal." Why are the results below being pre-empted here? Why not just go straight into it. I would remove these lines to save repetition.

Reviewer #3 (Remarks to the Author):

Overall comments:

Klitting et al explore the evolution of the Lassa virus in relation to its ecological region and predict potential for expansion beyond West Africa and this work is particularly relevant in the background of ongoing climate change and increased population and mobility across within Africa and across the globe.

The authors analysed geo-tagged genome sequences and environmental data This work is clearly written and very concise. The phylodynamic methods used in the analysis are well described.

The authors model the evolution of Lassa fever over a very wide timescale and over a large geographical region. I find this to be particularly challenging given the large of factors that can change a lot of the trajectories and parameters including improved healthcare, vaccines and economic growth across West Africa. Improved genome surveillance would be and will play a large role towards outbreak investigation and potential eradication of the virus over a wide time period. I would therefore constrain the projections to a shorter timeframe and that could be more useful to raise public health or mitigation measures to contain sporadic outbreaks.

Minor comments:

Line 203: it would be good to add a qualifying statement on the number of people that could avert the infections provided improved healthcare and outbreak response and given that circulation is remarkably slow in endemic areas.

Line. 219: ...“publicly available genome sequences (n=?)”

Line: 493: “ ... all publicly available sequences (n=?) – qualify with the date these were accessed.

Editor comments

Thank you again for submitting your manuscript "Predicting the evolution of Lassa Virus endemic area and population at risk over the next decades" to Nature Communications. We have now received reports from 3 reviewers and, after careful consideration, we have decided to invite a major revision of the manuscript.

As you will see from the reports copied below, the reviewers raise important concerns. We find that these concerns limit the strength of the study, and therefore we ask you to address them with additional work. Without substantial revisions, we will be unlikely to send the paper back to review. In particular, reviewer #2 has raised some concerns related to uncited literature that should be discussed in the context of your work and also questions the analysis techniques that were used to build the disease niche. This reviewer has comments suggesting the manuscript is split. We would prefer to keep the scope of the study the same but we would like you to address comments from this reviewer on the introduction and structure of your article so that the different aspects of the work are better linked together.

Please show all changes in the manuscript text file with track changes or colour highlighting. If you are unable to address specific reviewer requests or find any points invalid, please explain why in the point-by-point response.

*Answer: We would like to express our gratitude to the editorial team and the Reviewers for the constructive and relevant comments on the previous version of our manuscript. We have now revised the manuscript in response to the Reviewers' comments (see below for a point-by-point response to all comments). Additionally, we made some changes to make the format of the manuscript meet the *Nature Communications* recommendations. We hope this revised version is suitable for publication in *Nature Communications*.*

Reviewer 1 comments

This good and interesting paper about using niche modelling and viral phylogeography techniques to find the underlying driving risk factors for Lassa fever virus transmission in Africa, and then use forward projections of those drivers to estimate the changes to the endemic range of the virus. The paper is well written and uses recently developed techniques, which have also been used with other viral systems. It is also excellent that you have made your scripts and data (trees) available on Github.

Answer: We would like to thank the Reviewer for the positive assessment of our work.

I have only a few minor comments:

The main methods used were (a) niche models with boosted regression trees using reported occurrences of the virus and environmental factor data, and (b) phylogeographic models with viral sequence data and environmental factor data.

Both of these methods seem very suitable and have been used in similar studies before, either by the authors or others, but for different viruses (the references in the manuscript are sufficient). Niche models - as with most (all?) such studies disease occurrence data is used, and obviously if occurrences are under-reported then this will bias the results. Since the authors only have presence data, and not absence data they performed a suitable (and usual) background randomisation procedure (pseudo-absences) and only used spatial grid cells where there the main mouse host population was known.

Phylogeographic analyses - these were performed by first generating the time-scaled phylogenetic trees with spatial information then analysing the tree branch lengths and spatial diffusion paths with environmental rasters. The authors found that there were not any strong associations with the environmental predictors. This is only partly surprising, and as the authors explain could be due to the mobility of the main host species and generally slow dispersion rates of the virus.

What would be interesting to know though is - have those predictors changed through time from the temporal origin point (TMRCA) of the trees to the present ? Of course there might not be enough sampled sequences, temporal and spatial resolution to see whether this is an important effect or not, but it potentially might help explain the lack of signal (e.g. too averaged out in time or over long branch lengths). Therefore perhaps the authors could mention whether the predictors have substantially changed from say the 1900's or 1950's up until now in the discussion, and also mention (again) whether the predictors are expected to change at a higher or increasing rate from now into the future.

Answer: We fully share the Reviewer's concern on this point. Indeed, our procedure only considers constant-in-time environmental values that do not reflect the climatic and land cover conditions during the earliest part of Lassa virus dispersal history. To address this potential issue, we consider only the tip branches of our phylogenetic trees (Fig. S7) in the landscape phylogeographic analyses that aim at assessing the impact of environmental factors on the dispersal velocity of viral lineages. These tip branches correspond to the most recent dispersal events in the virus dispersal history and date back, on average, to the 1990s. In the revised version of the manuscript, in order to be more explicit and precise on this aspect, we have completed the section describing our approach as follows: "Our procedure only considers constant-in-time environmental values that do not reflect the climatic and land cover conditions during the earliest part of Lassa virus dispersal history, so we restricted our analyses to the most recent dispersal events (corresponding to tip branches of the trees from our phylogeographic reconstructions; Fig. S7). On average, this corresponds to considering lineage dispersal events starting in 1991 for segment L (95% HPD: [1988-1995]) and 1993 for segment S (95% HPD: [1991-1994])".

In addition, we also added a paragraph to the Discussion section detailing how some of our predictors changed since the 1900s and raising the issue of the impact of a possible acceleration

in environmental changes on our results: “Another caveat here is that our procedure considers constant-in-time environmental values that do not reflect changes in climatic and land cover conditions that may have occurred throughout the virus dispersal history. Indeed, it is estimated that in the past century alone, about 10 million ha of forest were lost in West Africa, mostly to the benefit of agricultural expansion [Norris et al. 2010]. Over the same period, annual precipitations have decreased throughout the area while temperatures have risen, up to over 1°C in some parts of Nigeria [Stocker et al. 2013]. To address this issue, we restricted our analysis to the most recent dispersal events in the virus dispersal history which date back, on average, to the 1990s. However, our approach still does not account for abrupt changes in climate, land use, and population density that may have occurred throughout the last decades in the West African region [Herrmann et al. 2020].”

Reviewer 2 comments

*This paper represents an analysis of the Lassa Fever and its host *Mastomys natalensis* in two parts. First from a 'niche modelling' approach where the authors examine the respective environmental niches of disease cases and known locations of the host. The second part is an analysis of sequence data where they examine the speed of spread of different variants of LASV.*

There is clearly a lot of careful and detailed work, backed by considerable expertise, that has gone into this paper. First, I note that a quick search of Lassa Fever quickly brings one to a very similar analysis by Redding et al. 2016 (<https://besjournals.onlinelibrary.wiley.com/doi/full/10.1111/2041-210X.12549>) that uses again uses niche-based methods to capture the disease case and host niches and projects them to the year 2070. This along with various papers by Moses et al. and a review by Gibb et al. (<https://pubmed.ncbi.nlm.nih.gov/28875769/>) need to be addressed to show how this paper is a progression from them. This goes beyond the request to cite some preferred papers; it is a small field so very similar papers and those more informational ones need to be cited.

Answer: We sincerely apologise for not having included particular references and discussion elements related to these studies on the ecological niche modelling of Lassa virus. We now refer to these studies, specifying how we add to these prior investigations (see below). While the study by Redding et al. (2016) also used ecological niche modelling to map the current risk of Lassa virus infections, they investigated the impact of "global change" from a different perspective, focusing on the quantification of spill-over events in the Lassa-endemic area. To perform this analysis, they used a different, environmental-mechanistic modelling approach that allowed them to incorporate sociological factors such as human mobility. Considering this paper and other studies by Peterson et al. (2014), Mylne et al. (2015), Fichet-Calvet et al. (2009), as well as the review by Gibb et al. (2017), our study complements previous works because it investigates the impact of projected environmental changes on the endemic range of the virus across the entire African continent. In contrast to the work by Redding and colleagues, we do not attempt to quantify spill-over events but investigate potential changes in the distribution of the virus. To fully address that question, we perform our analysis at the scale of the entire African continent, while Redding and colleagues carried out their assessment of future spill-over risk to West Africa. Furthermore, Redding and colleagues used an environmental-mechanistic model that allows them to include non-environmental factors such as human mobility. In our study, we combine two different strategies, namely, ecological niche modelling and phylogeography, to offer a comprehensive view of the potential drivers of Lassa virus occurrence and spread. In the revised version of the manuscript, we detail how our work adds to these previous studies and reference them.

The revised version of our manuscript has been completed as follows: "Ecological niche modelling studies have identified – although not always agreed on – environmental factors that correlate with the occurrence of Lassa virus infections in rodent and human hosts [Mylne et al. 2015, Redding et al. 2016, Peterson et al. 2014, Fichet-Calvet & Rogers 2009, Basinski et al. 2021]. Additional biological and socio-ecological factors may further influence spill-over dynamics, as suggested by mechanistic modelling investigations [Redding et al. 2016, Basinski et al. 2021, Iacono et al. 2016]. Most of these studies have mapped the current risk for Lassa virus infection in West Africa, identifying risk areas across much of this region [Mylne et al. 2015, Redding et al. 2016, Peterson et al. 2014, Fichet-Calvet & Rogers 2009, Basinski et al. 2021]. Like the rest of the world, African countries will increasingly be affected by climate change, with warming temperatures and more extreme, yet rarer, precipitation [Coumou & Rahmstorf 2012, Coumou et al. 2013, Bathiany et al. 2018]. These changes, combined with an increasing pressure on land resources due to a considerable projected human population expansion, are expected to result in important transformations of land use throughout Africa [Arneeth 2015, Brandt et al. 2017, Herrmann et al. 2020]. Previously, Redding and colleagues showed that spill-over events would at least double by 2070 within the Lassa-endemic western African region due to climate change, human population growth, and to a smaller extent, land use changes [Redding et al. 2016]. It is

not known however, how these environmental changes may affect the distribution of the virus itself [Gibb et al. 2017].

In this work, we add to previous modelling studies by investigating how the endemic range of Lassa virus may evolve in the next five decades in response to climate change, human population growth, and land use changes. [...]"

Overall, the structure of the paper is quite loose and seems to be a list of a wide variety of analyses rather than a tightly defined 'story' of the main findings and the sensitivity tests the authors have done to understand and back up these results. The introduction is somewhat vague and quite limited and does not seem to introduce the reader to all aspects of the varied analysis that are subsequently used, leaving readers from specific disciplines in the dark about what is being done and why.

Answer: We apologise that our study came across as a compilation of independent analyses to the Reviewer. The structure of the previous version of our manuscript aimed to follow the analytical flow we employed to investigate the present and future ecological niche and dispersal dynamic of Lassa virus. We believe that a collection of various analytical approaches, mainly ecological niche modelling/projections and (landscape) phylogeographic investigations, were required to get the most complete picture of the dynamics as well as potential drivers of Lassa virus occurrence and spread. To address the Reviewer's concerns, in the revised version of the manuscript, we have reworked the Introduction section to better contextualise and define the overall questions addressed in our study.

This work seems very much to be two separate manuscripts to me, loosely brought together, diverging around line 206. I would suggest nailing down the questions and constructing two careful and concise papers to address them.

One could be: How does the range of Mastomys natalensis alter over the next 50 years, given the variation we see in SSP/RCP scenarios? Without properly accounting for reporting biases in case data, any attempt to infer the 'niche' of the disease I believe is strongly flawed. After all what is the process causing there to be different disease case niche to the host niche? The authors mention it is not clear, so surely reporting biased is probably the most appropriate null expectation. Furthermore, any expansion of Mastomys natalensis range needs to also consider host dispersal speed and mechanisms (especially if you are going to do this for the virus) and the authors have demonstrated they have great expertise in e.g. diffusion models so this surely will be relatively straight-forward. I think any propose host range expansion would also need to consider edge effects in movement corridors but again, this should be fairly straight-forward. This work would need to build on the Redding et al. analysis, which doesn't seem too hard as there is much scope to expand here, to make a strong stand-alone paper.

Answer: We thank the Reviewer for this suggestion but as also indicated by the editor, we prefer maintaining and discussing the different analyses within a single study if at all possible. We believe that the different analytical approaches employed in our study lead to complementary results. For instance, while we do not find strong evidence that any of the environmental factors we tested influences the dispersal dynamics of viral lineages, our ecological niche modelling analyses do emphasise that environmental factors are important in determining the occurrence and hence local circulation of the virus. Furthermore, while our ecological niche modelling analyses indicate that the geographic area that is ecologically suitable to Lassa virus will likely expand in the upcoming years, our phylogeographic analyses confirm a relatively restricted dispersal capacity (i.e. a very low lineage dispersal velocity), which also allows us to illustrate through simulations that the virus might need time to spread through and occupy this extended suitable area. As mentioned in our previous answer, in the revised version of our manuscript, we have reworked our Introduction section to better explain the interest and complementarity of the different analytical approaches we perform.

The second main and very interesting finding is centred on the analysis of the sequence data, which as a separate analysis is both interesting and definitely adds to the current state of knowledge. The slow movement of Lassa relative to the small number of other data points the authors refer to is fascinating, and I think it can start to shed light on the limited geographic range of Lassa Fever in humans. I think a tightly defined and informative introduction, setting some clear questions and importantly introducing the reader to ideas that are examined e.g. phylogeography, how and why viruses spread etc. would be important here. I am not an expert on viral evolution, but having worked in organismal evolutionary biology I wonder how the relative fitness advantages of different virus strains impact these processes? If any mutations present have only a limited advantage i.e. the systems is some kind evolutionary equilibrium then the expectation would be of slow movement. In my naivety, some kind of key innovation would alternatively see a temporary faster movement will this mutation fixes in the population, before re-establishing some kind of (albeit brief) equilibrium. So isn't the state of the system important when measuring these velocities? If this is not the case it would be interesting to know why.

Answer: We agree with the Reviewer's comment that we cannot exclude the hypothesis that the emergence of a new strain/variant might have impacted the dispersal dynamics of the virus. For instance, we indeed estimated slightly different dispersal velocities for the three main clades analysed in our study (MRU clade, NGA clade II, and NGA clade III; Fig. S8). These differences are consistent between the two segments L and S, and could e.g. be due to differences in transmissibility impacting dispersal velocity and/or different sensitivities to environmental variations. While we do not find evidence for the latter hypothesis, we cannot exclude the first one, even if the difference between the different dispersal velocity estimates remains relatively small (all 95% HPD intervals are between 0.5 and 1.5 km/year). This aspect is now explicitly mentioned in the revised version of our manuscript: "*Consistently across the L and S segments, our estimates of the weighted lineage dispersal velocity for each of the main clades show that virus circulation is slowest for the MRU clade and fastest for the Nigerian clade II (Fig. S8). While we cannot exclude the hypothesis that one or several clades are associated with an increased transmissibility or shorter serial intervals accelerating virus dispersal, the generally low clade-specific estimates (< 1.5 km/year) remain similar.*"

The lack environmental signal perhaps could be to do with way that Mastomys natalensis is frequently moved around in agricultural produce, probably more in recent history due to increasing roads and mechanised transport. I am not sure if there is way to uncover whether there is more of a signal for older branches than more recent, i.e. there is temporal degradation over time, just a thought.

Answer: We thank the Reviewer for this interesting hypothesis. As detailed in one of our answers to Reviewer #1 (and now also further clarified in the text), we only considered the tip branches in our landscape phylogeographic analyses that aim at investigating the impact of environmental factors in the dispersal velocity of viral lineages. Indeed, because our procedure is based on the analysis of constant-in-time environmental values that do not reflect the climatic and land cover conditions during the earliest part of Lassa virus dispersal history, we restricted our analyses to the most recent dispersal events dating back, on average, to the 1990s (corresponding to tip branches of the trees from our phylogeographic reconstructions).

Furthermore, we assessed if it would be worth exploring the Reviewer's hypothesis of a lack of signal due to fast human-mediated dispersal events in the recent past. Briefly, we re-estimated the weighted dispersal velocity [and their 95% HPD interval] only associated with the tip branches corresponding to the more recent lineage dispersal events: 0.9 km/year [0.8-1.1] for segment L and of 1.0 km/year [0.9-1.2] for segment S. Those estimates are extremely close to the weighted lineage dispersal velocities estimated when considering all phylogenetic branches (0.8 km/year [0.7-1.0] for segment L, and 1.0 km/year [0.9-1.0] for segment S). If some fast, human-mediated, dispersal events did occur, we thus do not think that they had a notable impact on the overall dispersal velocity of viral lineages and hence on the results of our analyses dedicated to the environmental factors that might have impacted this velocity.

Some specific points:

Line 47: Need to expand and at least mention vaccine development and status.

Answer: We thank the Reviewer for suggesting this addition, we have now added details on vaccine development and status as well as antiviral drug availability and efficacy: “*There is currently no vaccine approved to prevent Lassa. Although several candidates have shown promising results during preclinical studies, only one (INO-4500) has progressed to clinical trials (now in phase 1B, NCT03805984) [Purushotham et al. 2019, Mateo et al. 2021]. Regarding Lassa treatment, the only antiviral drug available is the nucleoside analog ribavirin [McCormick et al. 1986], which is often ineffective [Eberhardt et al. 2019].*”

Line 50: Also very brief; needs to mention huge impact of historical under-reporting and biases in the data.

Answer: We thank the Reviewer for suggesting this addition, this aspect is now explicitly addressed in the text (see also our detailed answer to a related comment below).

Line 57: Where is the reference for this statement, hugely unknown mechanism of spill-over, needs altering or strongly qualifying?

Answer: We thank the Reviewer for pointing out the need for rephrasing and further referencing the statement at line 57 (“*Most infections occur through exposure to the excreta of infected *Mastomys natalensis**”). We think that the status of *M. natalensis* as the main reservoir of Lassa virus is now well-established, as is the importance of spill-overs in contributing to human infections. Indeed, following the initial identification of Lassa virus reservoir by Monath *et al.* in 1974 (PMID: 4833828), studies of rodent biology, ecology, transmission dynamics, and viral genomes, demonstrated that most human infections are acquired from rodents (PMIDs: 11987809, 17326956, 33564902, 17378212, 9025695, 25569707, 30332564, 30606844, 26276630). That said, we do agree that the mechanism of spill-over itself remains to be formally established as humans may be infected either through direct contact (notably via bushmeat hunting; PMIDs: 9025695, 9025695) or indirectly, by exposure to infected rodent excreta, as suggested by observational and ecological studies (PMIDs: 28167603, 28167603), *in vivo* infections rodents (PMIDs: 34792436, 34792436, 28438110) and non-human primates (PMID: 32485952), assessment of Lassa virus stability in aerosols (PMID: 6512508), as well as by analogy with other arenaviruses (PMIDs: 1371270, 26139838). Accordingly, we corrected and better referenced our statement in the main text: “*Human infections are generally thought to occur through direct contact or exposure to the excreta of infected *Mastomys natalensis* [Monath et al. 1974, Stephenson et al. 1984, Wozniak et al. 2021, Ter Meulen et al. 199, Downs et al. 2020], although the main transmission mechanism remains to be definitively determined.*”

Line 70-74: Need to go into more details about the climate links e.g. referencing the large body of work by Fitchet-Calvet and colleagues.

Answer: We thank the Reviewer for suggesting this addition, we now refer to studies by Fichet-Calvet and colleagues regarding the environmental drivers of Lassa virus transmission in the Introduction section.

Line 90: why are we going back into introduction after a heading saying ‘Results’?

Answer: We have now re-organised the text to avoid the inclusion of Introduction elements in the Results section.

Line 98: The data are so biased here it would be stretch to say you are able to infer the actual disease niche. The difference here between the host and disease niche, is likely to be strongly influenced by the signal of disease surveillance. To account for this, the authors need to use a modelling approach such as hierarchical mixed models that you can better use to understand the

system, not mainly for making predictions as is usually the case with BRTs. To draw any conclusions, one would need to think causally about the process that driven the patterns of cases seen and attempt to control for those unmeasured confounding factors, such as health-seeking behavior and availability of diagnostic tests. For instance, Nigeria has the highest capacity and is also the area where most of the data come from.

Answer: We agree with the Reviewer that the heterogeneous disease surveillance effort has biased the spatial distribution of Lassa records. However, we argue that its impact should here be rather limited for the following reasons: (i) our BRT models have been trained using presence and pseudo-absence points while considering a gridded space associated with a low resolution. The necessity to work on such a low resolution (grid cells of 0.5x0.5 decimal degrees, i.e. a resolution of 30 arcmin) was dictated by the resolution of environmental rasters available to perform future projections. As described in our Methods section, “we only sampled pseudo-absences in raster cells in which the presence of at least one individual of another species belonging to the Muridae family has been recorded. Because it only requires a single occurrence record to consider a presence, we discarded all but one occurrence record per raster cell. We applied the same filtering step for the pseudo-absence points and simply discarded pseudo-absences falling in raster cells with occurrence data.” Therefore, even in areas where the disease surveillance was limited, it only necessitated at least a single occurrence record to consider a Lassa virus presence in the large grid cell we use, which decreases, yet not totally avoids, the risk of artefacts due to spatially heterogeneous surveillance effort in our BRT models. (ii) Moreover, pseudo-absence points were only sampled in grid cells within which the presence of the host has been confirmed by occurrence records, a procedure followed to “avoid treating under-sampled areas as ecologically unsuitable for the virus, but also to account for potential heterogeneity in sampling effort or surveillance [Phillips et al. 2009, Elith et al. 2011]”. While (ii) was already detailed in our Methods section, we now also explicitly mention (i) in the same section: “This filtering procedure is required to only have a presence or a pseudo-absence assigned to each grid cell, but also has the advantage of minimising the risk of artefacts due to spatially heterogeneous disease/wildlife surveillance efforts in the resulting BRT models. Indeed, even in areas where the disease surveillance was limited, it only necessitates at least a single occurrence record to consider the presence of the target species in the large grid cell we use.”

Line 112: I am not sure these are the best methods for this purpose. Also, this all seems like methods rather than results.

Answer: Because the Methods section is at the end, we here only include an introductory statement to mention the general methodology employed to generate the results presented in that subsection of the manuscript. Regarding the choice of the methodology, ecological niche modelling approaches constitute relevant and well-established tools to investigate the relationship between environmental factors and the ecological suitability of a given species (sometimes interpreted as the probability to observe the target organism). Among the different methodologies available, the boosted regression trees (BRT) approach has been identified as one of the most performant methods (Elith et al. 2006, doi: 10.1111/j.2006.0906-7590.04596.x).

Line 191: Why this number – what influence do other numbers have?

Answer: Although this choice is arbitrary (we chose 0.5 because the ecological suitability values range between 0 and 1), it allows comparing the different time periods with the same cut-off. If we increase or decrease that cut-off value, we basically find the same trends when comparing the evolution of the population at risk of living in a Lassa virus endemic area (see the newly extended Table S1). Regardless of the cut-off value used (0.25, 0.50 or 0.75), we find that the population living in areas suitable for Lassa virus will increase by a factor ~3 in 2030, by a factor ~5 in 2050, and by a factor ~8 in 2070. In addition, while absolute numbers change, they remain on the same order of magnitude, with current population estimates of 124, 92, and 51 for the current time to estimates of 1011, 700, and 393 million in 2070 for cut-offs of 0.25, 0.5, and 0.75, respectively (Table S1). We now explicitly mention this sensitivity test in the text. We have also edited the text

to avoid putting too much emphasis on the resulting absolute estimates of the number of people at risk, but more on the trends.

“More specifically, our results show that by 2070, 363 [333-384] million people may be exposed to Lassa virus infection in current suitable areas and that expansion of the ecological niche of the virus might put 337 [260-405] million more people at risk of infection. As our estimates may be sensitive to the arbitrary cut-off value of 0.5 we used to define an area as ecologically “suitable”, we repeated our analysis considering different arbitrary cut-off values (0.25 and 0.75), and observed the same trends (Table S1).”

Lines 193: This seems very low, again what is role of underreporting here?

Answer: The absolute values reported here depend on the cut-off of ecological suitability used for those computations. We have edited the text to be more explicit about this aspect. For the potential impact of reporting, we refer to our reply to the earlier comment (and related edits) on the heterogeneous disease surveillance effort.

*Lines 200-204: I am very dubious about these numbers, it all stems from believing the LASV niche model and whether the host (or hosts) would actually spread, followed by the virus. Species tend to move relatively slowly with climate change (<https://www.science.org/doi/10.1126/science.1206432>) so how far would *Mastomys natalensis* have spread in this time? How does the boom-bust population dynamics of *Mastomys natalensis* look in these new environments, and how important are these population processes.*

Answer: Our ecological niche modelling analyses indicate that the distribution of the main host species (*M. natalensis*) should remain rather stable over the next decades (Figs. 2 and S1). Therefore, we explicitly base our discussion on the potential spatial extension of the ecological niche of the virus - this aspect is now explicitly stated in the Results section: *“Our projections show that the potential expansion of the ecological niche of Lassa virus would occur in regions where *M. natalensis* is (and will remain) present, as indicated by our estimates of present and future suitability for *M. natalensis* across Africa ([Mylné et al. 2015, Redding et al. 2016, Peterson et al. 2014, Fichet-Calvet & Rogers 2009] and Fig. 2). In the next decades, with *M. natalensis* being present where ecological conditions will be suitable for virus circulation outside of the current endemic range, the distribution of Lassa virus in Africa could potentially widen. The recent emergence of Ebola virus in West Africa and of West Nile virus in North America illustrate how zoonotic viruses can travel over long distances to effectively settle in new regions [Iacono et al. 2016, Coumou & Rahmstorf 2012, Coumou et al. 2013, Bathiany et al. 2018], dramatically widening their range, and putting more people at risk of infection”*. Moreover, a spatial extension of the niche does not necessarily imply an actual spatial extension of the virus distribution. We therefore exclusively talk about *“people at risk of living in areas [ecologically] suitable for Lassa virus.”* Finally, as answered above, we agree that the absolute numbers we report should be interpreted with the ecological suitability value used as cut-off in mind.

Lines 275-279: “major waterways may have limited impact on virus dispersal.” Why are the results below being pre-empted here? Why not just go straight into it. I would remove these lines to save repetition.

Answer: Thank you for the suggestion, we have now removed those sentences.

Reviewer 3 comments

Overall comments:

Clitting et al explore the evolution of the Lassa virus in relation to its ecological region and predict potential for expansion beyond West Africa and this work is particularly relevant in the background of ongoing climate change and increased population and mobility across within Africa and across the globe.

The authors analysed geo-tagged genome sequences and environmental data This work is clearly written and very concise. The phylodynamic methods used in the analysis are well described.

Answer: We thank the Reviewer for the positive feedback on our work.

The authors model the evolution of Lassa fever over a very wide timescale and over a large geographical region. I find this to be particularly challenging given the large of factors that can change a lot of the trajectories and parameters including improved healthcare, vaccines and economic growth across West Africa. Improved genome surveillance would be and will play a large role towards outbreak investigation and potential eradication of the virus over a wide time period. I would therefore constrain the projections to a shorter timeframe and that could be more useful to raise public health or mitigation measures to contain sporadic outbreaks.

Answer: We understand the concern raised by the Reviewer but we specifically wanted to consider short (2030), middle (2050), as well as long (2070) term perspectives. In all cases, we can see that the potential increase in the number of people in areas ecologically suitable to Lassa virus would be mostly driven by the projected short term increase in local population sizes rather than by the geographical extension of this area (Fig. 2B). Moreover, as stated in our Discussion, the virus would have to spread over relatively long distances to reach new ecologically suitable regions in Central and East Africa. For these reasons, we mainly discuss the impact of population growth in already endemic areas. However, in the previous version of our Discussion, the emphasis was indeed mainly put on the long-term perspectives, i.e. the projection for 2070. We have now addressed this issue by reporting and comparing the projections for 2030 and 2070: “When considering a cut-off value of 0.5 for the suitability defining the niche of the virus, we estimated that population growth in endemic countries could alone put 186 (95% HPD interval: [172-196]) and 341 [315-369] million people at risk of infection by 2030 and 2070, respectively (Table S1), compared to an estimated 83 [76-87] million today.”

We also corrected some numbers that were incorrectly reported from Table S1 to the main text (these changes have no impact on our conclusions). In detail, 453 [414-498] was changed to 453 [414-497] in the Results section; 341 [308-360] was corrected to 341 [315-369] and 92 [83-98] was changed to 83 [76-87] in the Discussion section.

Minor comments:

Line 203: it would be good to add a qualifying statement on the number of people that could avert the infections provided improved healthcare and outbreak response and given that circulation is remarkably slow in endemic areas.

Answer: We thank the Reviewer for this insightful suggestion. We fully agree that improved healthcare and outbreak response – including heightened surveillance, improved sanitation and increased awareness – should significantly alleviate the burden of Lassa fever. It is, however, challenging to evaluate the number of infections that could be prevented using the mitigation efforts cited above. We added a paragraph in the Discussion section, highlighting that these measures may significantly decrease exposure to Lassa virus infection: “In this analysis, we focused on evaluating the population that will likely be exposed to Lassa in endemic regions or that might be exposed in regions that we predict to be suitable for Lassa virus in the future. Importantly, we did not evaluate how much heightened surveillance, improved sanitation and

increased awareness may help to reduce exposure to Lassa virus in the future [Buba et al. 2018, Tobin et al. 2015, Saez et al. 2018, Ejembi et al. 2019]. These measures will, however, remain crucial to prevent infections. Even more so as we report an extremely slow dispersal velocity of Lassa virus in endemic areas, suggesting that localised efforts that target infection hotspots may prove to be highly effective.”

Line. 219: ... “publicly available genome sequences (n=?)”.

Answer: We thank the Reviewer for highlighting this issue and have now specified in that paragraph the number of sequences analysed for each segment.

Line: 493: “... all publicly available sequences (n=?) – qualify with the date these were accessed.

Answer: We have now added detailed L and S segment sequence numbers, and access date for Lassa sequences downloaded from the NCBI Nucleotide database.

Additional changes made by the authors

During the revision process, we have also included and added two additional co-authors (Sylvanus Okogbenin and Ephraim Ogbaini-Emovo) who were actively involved in the sampling process of the new 2019 Lassa virus samples whose genomes were sequenced and analysed in the present study.

In order to follow GBIF recommended citation format, we incorporated download DOIs into the relevant method section: “*We obtained occurrence data for *M. natalensis* species from the Global Biodiversity Information Facility (GBIF.org, <https://doi.org/10.15468/dl.hrjyj1>, accessed 2019-07-19). [...]. Occurrence data for the Muridae family were obtained from the GBIF database (GBIF.org, <https://doi.org/10.15468/dl.cs3c41>, accessed 2019-07-19).”*

REVIEWER COMMENTS

Reviewer #1 (Remarks to the Author):

Thank you for considering and answering my comments in the first round of review. Also in this round of review, I checked the points raised by reviewer 2 and have considered your revisions in light of these as well.

In reply to the points raised - I have a few comments, but no more suggested revisions, because I think you have now addressed the points raised in the first round of revisions suitably.

Responses to Reviewer 1 -

Specifically, my point about time varying predictors, to which you reply that you are only using "constant-in-time environmental values", but recognising that this is not exactly ideal you say "to address this potential issue, we consider only the tip branches of our phylogenetic trees". I think that considering only the recent branches (which you do) does indeed help with the problem and at least stops the inference happening over the long early branches. So I think this is OK, especially since you then go on to include the when the dispersal events that are being modelling start from in time.

I see that you have also now included a part in the discussion section which caveats the results, since abrupt changes in driver conditions are not included, which is OK too.

Nevertheless (but not for this current study), I do think it telling that you did not find very significant drivers, and I suspect that not including changing underlying drivers could be an issue here. However I'm not suggesting any further revisions on this point because to do that would require writing a new software package which is really beyond the scope of this paper (and I'm not aware of other phylodynamic packages to do something similar that would be useable).

Responses to Reviewer 2 -

R2.1 Reviewer 2 advises to include some other Lassa fever studies, which you do, and your answer and the revisions look suitable. One point though, Reviewer 2 says 'very similar analysis by Redding et al' - which I don't entirely agree with, the although the Redding et al paper is about predicting and projecting region / niches of zoonotic spill over events, the phylogenetic analyses in that paper is of a host species, which is really quite different methodology to the phylodynamic analyses of the virus in this manuscript.

R2.2 (Structure of paper particularly the Introduction) - Reviewer 2 suggests some more clarity on the paper structure, and I think the problem originally was that the structure had the (detailed) Methods at the end. I see you have included an outline of the different methods at the end of the Introduction, which I think is now OK.

R2.3 (Two separate analyses) - I found it interesting that the two types of analyses: Boosted Regression Trees (spatial cases) and Phylodynamics/phylogeography of the virus genomes were both used to complement each other in the same paper.

R2.4 (Viral phylogeography & dispersal speeds especially of new variants) - The question from Reviewer 2 is about whether a new strain or variant might spread faster in the population, and your response is that only slight differences between viral clades were detected anyway. Therefore I think the revision to the manuscript is suitable.

R2.5 (Lack of environmental signal) - Reviewer 2 also asks about lack of environmental signal in the phylogeographic GLM analysis, and you reply that the recent tip branches were considered. I see you also note that when only very recent branches in the tree are considered that the dispersal velocity in these was also not significantly different to that presented in the manuscript (from approx 1990s onwards). Therefore I agree that just using the branches from 1990s onwards seems good enough in this case, although note my comments above.

Reviewer #3 (Remarks to the Author):

I am glad that the authors have addressed the concerns that I raised in addition to those raised by other two reviewers. The revised work reads well. I am happy with the revisions and no further comments.

Reviewer #4 (Remarks to the Author):

comments file uploaded

Predicting the evolution of the Lassa virus endemic area and population at risk over the next decades – review of the work by Klitting *et al.* and of their responses to reviewers' suggestions.

The authors have conducted a major study analysing the predictors of Lassa fever virus endemicity and spread, both now and extrapolating into the future (up to 2070). They have used separate methods to consider the environmental distribution of the disease and its vector and how (fast) the virus may spread in space and time.

This is a huge amount of work on a most important virus and rather than going through commending the elements that are very interesting, I focus here on the issues (major and minor) that I think are troubling with the approach taken.

Major issues

- A) Lines 60-68 – given the recognition of the importance of rodent host phylogroup given here, clearly demonstrated in Redding's work, why do the authors simply ignore this in their predictions? The environmental niches that are filled by phylogroups not currently transmitting Lassa fever virus mean that one cannot simply assume that the virus carrying rodents will be able to expand into these areas over time, without an impact of competition.
- B) Concerning the ecological niche modelling done, I was not really convinced by the arguments used by the authors that they had done something as novel as they claim – what they seem to have done is to follow similar methods to Redding et al, but ignore the phylogroup of *Mastomys natalensis*. and to extrapolate across the continent, away from the virus training dataset (necessarily restricted to the current endemic area of Lassa). Without getting into the details of the extrapolation reliability of the specific methods used (and most workers feel that extrapolation outside of predictions are unreliable, whether using BRT or even INLA) the geographical extent of this, which ignores the complication of the possibility of vector phylogroup impact, above, makes this approach highly unreliable. It is remarkable to me that they were not aware of or had not referred to Redding's work in earlier versions and their response for me is troubling rather than reassuring.
- C) The authors assume that virus distribution is determined or limited by *Mastomys natalensis* distribution. It was surprising to me that Elisabeth Fichet Calvet's results that other species, not least *M. erythroleucus* being important elsewhere in West Africa (and potentially therefore elsewhere) was being overlooked in this work. This may, contrary to above, relate to the suggestion that savannah impedes transmission of Lassa, as *M. erythroleucus* favours such an ecosystem.
- D) Building on A & B above, the extrapolation in distance (and the logic used here) and (extended periods of) time the authors seem to be assuming that *M. natalensis* and Lassa viruses are expanding into naïve empty systems; the reality is that there are other species of rodent carrying other arenaviruses across sub-Saharan Africa – there might already be competent vectors for Lassa fever elsewhere in the continent but no ecological niche for the virus as something similar is there already. The methods used cannot really address this question – and there is no reason to assume that Lassa will expand the other arenaviruses decline, rather than the other arenaviruses expand into the Lassa region, replacing Lassa fever virus. This relates to the major concern raised by referee 3, which I do not feel that the authors have really addressed.
- E) Uncertainty is not reflected in any of the prediction maps.

In summary, I do not feel that it is sensible to extrapolate predictions from the current Lassa fever virus distribution across the whole sub-continent. This is not just a question of the methods used – but rather relates to the assumptions of expanding into already full ecological niches that simply may not have ‘space’ for Lassa to expand into.

Minor issues

L54 – the cases in Ghana have never been confirmed – and indeed one of this manuscript’s authors told me that their further investigations suggested that these cases were not real, so why are they listed here?

L180-182 – the epidemic spread of Ebola in humans, whose spillover related distribution is usually restricted by the endemic dynamics in animal reservoirs bears no relation to the distribution of Lassa, determined by vector dynamics. Further, the translocation of WNV to US from Israel into a novel ecosystem is a qualitatively different consideration to the endemic spread of Lassa regions.

L270 and Table 1. This table is comparing ‘apples and pears’ and is not valuable, but rather I feel is misleading. Spread in epidemic systems in naïve ecological niches (especially human Ebola in West Africa) involves totally different routes etc compared to endemic Lassa transmitting in animal vectors. Similarly many of the rabies transmission rates describe the virus transmitting into naïve populations – something rather different when considering how Lassa needs to compete with other Lassa lineages or other arenaviruses.

We would like to thank all the Reviewers for their constructive comments on the revised version of our manuscript. We have provided a point-by-point response to all Reviewer's comments and revised the manuscript accordingly.

Reviewer 1 comments

Thank you for considering and answering my comments in the first round of review. Also in this round of review, I checked the points raised by reviewer 2 and have considered your revisions in light of these as well.

In reply to the points raised - I have a few comments, but no more suggested revisions, because I think you have now addressed the points raised in the first round of revisions suitably.

Responses to Reviewer 1 -

Specifically, my point about time varying predictors, to which you reply that you are only using “constant-in-time environmental values”, but recognising that this is not exactly ideal you say “to address this potential issue, we consider only the tip branches of our phylogenetic trees”. I think that considering only the recent branches (which you do) does indeed help with the problem and at least stops the inference happening over the long early branches. So I think this is OK, especially since you then go on to include the when the dispersal events that are being modelling start from in time.

*Answer: We thank the Reviewer for going through our responses and revised manuscript. We are glad that the reviewer is satisfied with our approach to limit possible biases due to the use of constant-in-time environment values in our *post hoc* phylogeographic analyses.*

I see that you have also now included a part in the discussion section which caveats the results, since abrupt changes in driver conditions are not included, which is OK too.

Nevertheless (but not for this current study), I do think it telling that you did not find very significant drivers, and I suspect that not including changing underlying drivers could be an issue here. However I'm not suggesting any further revisions on this point because to do that would require writing a new software package which is really beyond the scope of this paper (and I'm not aware of other phylodynamic packages to do something similar that would be useable).

*Answer: We agree with the Reviewer that incorporating time-variations while assessing the impact of environmental factors on the dispersal velocity of viral lineages would be interesting. As developed in our previous rebuttal letter, we believe that limiting our analyses to dispersal events dating back to on average 30 years ago reduces considerably the risk of overlooking a possible impact of environmental factors on virus dispersal. In addition and as raised by the reviewer, incorporating time varying variables in our analysis would indeed require developing a novel methodological approach. More specifically, and as outlined by Dellicour *et al.* (PMID: 26864798), the analysis of time-varying environmental factors should be theoretically feasible, but poses significant technical and practical problems like the generalisation of the least-cost and/or Circuitscape path models to three rather than two dimensions, with the third dimension corresponding to rasters that represent different points in time. We are hopeful that the present study will serve as a starting point for such developments and analyses.*

*R2.1 Reviewer 2 advises to include some other Lassa fever studies, which you do, and your answer and the revisions look suitable. One point though, Reviewer 2 says ‘very similar analysis by Redding *et al*’ - which I don't entirely agree with, the although the Redding *et al* paper is about predicting and projecting region / niches of zoonotic spill over events, the phylogenetic analyses in that paper is of a host species, which is really quite different methodology to the phylodynamic analyses of the virus in this manuscript.*

*Answer: We are glad that the Reviewer is satisfied with our revisions in response to Reviewer #2's comments. We agree that our study differs from the work of Redding and colleagues. They perform a phylogeographic analysis of the *M. natalensis* species to identify its main subclades, while we perform phylogeographic analyses to investigate a potential effect of environmental factors on the dispersal dynamic of the virus. In addition, our ecological niche modelling*

analyses also differ from what has been previously done by Redding and colleagues. As we highlight in our previous point-by-point response, we focus our analyses on the ecological suitability for Lassa virus circulation while Redding and colleagues estimate the number and spatial extent of Lassa virus spill-over events.

R2.2 (Structure of paper particularly the Introduction) - Reviewer 2 suggests some more clarity on the paper structure, and I think the problem originally was that the structure had the (detailed) Methods at the end. I see you have included an outline of the different methods at the end of the Introduction, which I think is now OK.

Answer: We are glad that the Reviewer is satisfied with our revisions addressing Reviewer #2's concerns regarding the structure of the manuscript.

R2.3 (Two separate analyses) - I found it interesting that the two types of analyses: Boosted Regression Trees (spatial cases) and Phylodynamics/phylogeography of the virus genomes were both used to complement each other in the same paper.

Answer: We thank the Reviewer for this positive assessment of our choice of analytical approaches.

R2.4 (Viral phylogeography & dispersal speeds especially of new variants) - The question from Reviewer 2 is about whether a new strain or variant might spread faster in the population, and your response is that only slight differences between viral clades were detected anyway. Therefore I think the revision to the manuscript is suitable.

Answer: We are glad the Reviewer finds that our revisions in response to Reviewer #2's comments are acceptable.

R2.5 (Lack of environmental signal) - Reviewer 2 also asks about lack of environmental signal in the phylogeographic GLM analysis, and you reply that the recent tip branches were considered. I see you also note that when only very recent branches in the tree are considered that the dispersal velocity in these was also not significantly different to that presented in the manuscript (from approx 1990s onwards). Therefore I agree that just using the branches from 1990s onwards seems good enough in this case, although note my comments above.

Answer: We are glad that the Reviewer is satisfied with our approach to limit possible biases due to the use of constant-in-time environment values in our *post hoc* phylogeographic analyses.

Reviewer 3 comments

I am glad that the authors have addressed the concerns that I raised in addition to those raised by other two reviewers. The revised work reads well. I am happy with the revisions and no further comments.

Answer: We thank the Reviewer for their positive assessment of our work.

Reviewer 4 comments

The authors have conducted a major study analysing the predictors of Lassa fever virus endemicity and spread, both now and extrapolating into the future (up to 2070). They have used separate methods to consider the environmental distribution of the disease and its vector and how (fast) the virus may spread in space and time. This is a huge amount of work on a most important virus and rather than going through commending the elements that are very interesting, I focus here on the issues (major and minor) that I think are troubling with the approach taken.

Major issues:

A) Lines 60-68 – given the recognition of the importance of rodent host phylogroup given here, clearly demonstrated in Redding’s work, why do the authors simply ignore this in their predictions? The environmental niches that are filled by phylogroups not currently transmitting Lassa fever virus mean that one cannot simply assume that the virus carrying rodents will be able to expand into these areas over time, without an impact of competition.

*Answer: We agree with the Reviewer that we cannot assume that one *M. natalensis* phylogroup may freely expand in an area where closely related phylogroups are present, as they may occupy similar ecological niches. This aspect is, however, outside the scope of our analyses. Our projections are focused on future changes in ecological suitability for Lassa virus circulation. We do not project changes in the distribution of the virus itself, but investigate scenarios of spatial extension of its ecological niche that could potentially result in the introduction of the virus in new ecologically suitable areas. In doing so, we assume that the virus would be able to spread locally as (i) environmental conditions are suitable for virus circulation and (ii) a suitable host, *M. natalensis*, is present (since the species is distributed throughout Sub-Saharan Africa with the exception of South Africa). As detailed in the Discussion section of our manuscript, while some studies have postulated that Lassa virus circulation may be limited to *M. natalensis* phylogroup A-I, records of Lassa virus infection in *M. natalensis* phylogroup A-II and in other rodent species including *M. erythroleucus* or *Hylomyscus pamfi* suggest that susceptibility to Lassa virus infection may not be species- or phylogroup- specific. For this reason, we consider the entire *M. natalensis* group to be *a priori* a suitable host for Lassa virus.*

To address the Reviewer’s concerns, we further clarify this assumption and discuss the possible ensuing limitations in the Discussion section:

*“In addition, some factors may hinder virus dispersal following a hypothetical introduction into a new ecologically suitable region. First, in our analysis, as there is evidence that Lassa virus infection may not be species or subtaxon specific [92-93], we considered the entire *M. natalensis* species to be susceptible to Lassa virus infection. It is, however, possible that the virus spreads less (or more) efficiently in a different subtaxon of *M. natalensis*.”*

*B). Concerning the ecological niche modelling done, I was not really convinced by the arguments used by the authors that they had done something as novel as they claim – what they seem to have done is to follow similar methods to Redding et al, but ignore the phylogroup of *Mastomys natalensis*. and to extrapolate across the continent, away from the virus training dataset (necessarily restricted to the current endemic area of Lassa). Without getting into the details of the extrapolation reliability of the specific methods used (and most workers feel that extrapolation outside of predictions are unreliable, whether using BRT or even INLA) the geographical extent of this, which ignores the complication of the possibility of vector phylogroup impact, above, makes this approach highly unreliable. It is remarkable to me that they were not aware of or had not referred to Redding’s work in earlier versions and their response for me is troubling rather than reassuring.*

Answer: We apologise for not having included a reference to the study from Redding and colleagues in the initial version of our manuscript. It was not our intention to claim that our

ecological niche modelling approach was novel in itself, and we apologise if the reviewer got this impression. We believe the novelty of our work lies in the fact we combine analytical approaches, mainly ecological niche modelling/projections and (landscape) phylogeographic investigations, to explore how the endemic range of Lassa may evolve in the next five decades. We think this combination was required to get the most complete picture of the dynamics as well as potential drivers of Lassa virus occurrence and spread – a view shared by Reviewer 1. As also detailed in our third response to Reviewer 1 above, our study differs from the work of Redding and colleagues, because (i) we focus our ecological niche modelling analyses on the ecological suitability for Lassa virus circulation while Redding and colleagues estimate the number and spatial extent of Lassa virus spill-over events; (ii) in the phylogeographic part of our study, we use different yet complementary analytical approaches and address different questions. Redding and colleagues perform a phylogeographic analysis of the host species (*M. natalensis*) to identify its main subclades, while we conduct landscape phylogeographic analyses to investigate a potential impact of a series of environmental factors on the dispersal dynamic of the virus.

As detailed in our Methods section, our ecological niche model for the Lassa virus is not only trained on its current endemic area, but on the entire study area corresponding to the current *M. natalensis* range: “For Lassa virus, we only sampled pseudo-absences in raster cells in which the presence of *M. natalensis* has been recorded. While heterogeneous disease surveillance efforts likely bias the spatial distribution of Lassa records [48,49], this procedure avoids treating under-sampled areas as ecologically unsuitable for the virus, but also limits the potential impact of such heterogeneity in sampling effort or surveillance [110,111].” Technically, we thus do not extrapolate but explore the evolution of areas ecologically suitable for the virus within the entire host species range. As detailed in the response above, we indeed consider the entire *M. natalensis* species as suitable for Lassa virus and for that reason, did not account for competition between phylogroups. We refer to our previous reply on how we address this concern in the Discussion section.

C). *The authors assume that virus distribution is determined or limited by Mastomys natalensis distribution. It was surprising to me that Elisabeth Fichet Calvet’s results that other species, not least M. erythroleucus being important elsewhere in West Africa (and potentially therefore elsewhere) was being overlooked in this work. This may, contrary to above, relate to the suggestion that savannah impedes transmission of Lassa, as M. erythroleucus favours such an ecosystem.*

Answer: We fully agree that, based on the work of Fichet-Calvet and others, *M. erythroleucus* may be important for Lassa virus maintenance elsewhere in West Africa (and in Africa in general). In our ecological niche modelling analyses, we chose to be conservative and focused on the main species recognised as a reservoir host for Lassa virus to define our background, an aspect that we now more explicitly emphasised in the text. Our results suggesting that savannas may slow down the circulation of Lassa virus are, however, completely independent from any assumption regarding the host as they were obtained through phylogeographic analyses based on all viral sequences available for Lassa virus, irrespective of the host in which the sequence was collected.

“Of note, while several rodent species such as *M. erythroleucus* [93] may be important for Lassa virus transmission, we chose to be conservative and focused on the main species recognised as a reservoir host for Lassa virus to define our background.”

D). *Building on A & B above, the extrapolation in distance (and the logic used here) and (extended periods of) time the authors seem to be assuming that M. natalensis and Lassa viruses are expanding into naïve empty systems; the reality is that there are other species of rodent carrying other arenaviruses across sub-Saharan Africa – there might already be competent vectors for Lassa fever elsewhere in the continent but no ecological niche for the*

virus as something similar is there already. The methods used cannot really address this question – and there is no reason to assume that Lassa will expand the other arenaviruses decline, rather than the other arenaviruses expand into the Lassa region, replacing Lassa fever virus. This relates to the major concern raised by referee 3, which I do not feel that the authors have really addressed.

Answer: We fully understand the concerns of the Reviewer regarding a potential competition between Lassa virus and other arenaviruses but this aspect is out of the scope of our analyses. As this aspect is relevant to the more general question of the expansion of the range of Lassa virus, it was addressed in the Discussion section of the initial and revised versions of our manuscript. To respond to the Reviewer's concerns, we now further insist on how potential competition with other viruses could potentially affect the spread of Lassa virus in a new ecologically suitable area in an additional paragraph included in the Discussion section:

“In our study, we use phylogeographic simulations to highlight how, in the absence of restrictions from the environment, a slow lineage dispersal velocity may limit the propagation of Lassa virus in case of a successful introduction into a new ecologically suitable area. We use these simulations for illustration and not prediction as the dispersal dynamics upon virus emergence in a new region are unclear. The virus may spread swiftly through an immunologically naïve rodent population, but the low mobility of the rodent reservoir could still limit the velocity of virus dispersal on a larger scale. As underlined above, a number of other elements may also come into play, such as the nature of host species or subtaxa as well as potential cross-immunity or competition due to the local co-circulation of closely related viruses.”

*“In addition, some factors may hinder virus dispersal following a hypothetical introduction into a new ecologically suitable region. First, in our analyses, as there is evidence that Lassa virus infection may not be species or subtaxon specific [92,93], we considered the entire *M. natalensis* species to be susceptible to Lassa virus infection. It is, however, possible that the virus spreads less (or more) efficiently in a different subtaxon of *M. natalensis*. Second, while in most of the areas where ecological suitability for LASV appears to increase over time outside of West Africa (e.g. DRC, Uganda, Cameroon), no Mammarenavirus species have so far been identified [13], we cannot rule out the possibility that potential cross-immunity or competition due to the local co-circulation of closely related viruses may hinder the propagation of LASV.”*

E). Uncertainty is not reflected in any of the prediction maps.

Answer: We apologise for not making this point more obvious in the Results section. The uncertainty in our predictions is shown in the supplementary data (**Fig. S1**), where we report the standard deviations measured over the four different climatic models used for our projections. We have now modified the text to clarify this point.

In the Results section: *“RCP 2.6 and RCP 8.5 are the most extreme scenarios and refer to either stringent mitigation (RCP 2.6), or high-end emissions (RCP 8.5), while RCP 6.0 represents a medium-high emission scenario [61]. Focusing on RCP 6.0, we projected that by 2070, most of the region between Guinea and Nigeria will become suitable (ecological suitability >0.5) for Lassa virus (Fig. 2A, see Fig. S1 for the other scenarios as well as for the standard deviations associated with all projections)”*.

In the legend of Figure 2: *“see Figure S1 for the other scenarios as well as for the standard deviations associated with all projections”*.

In summary, I do not feel that it is sensible to extrapolate predictions from the current Lassa fever virus distribution across the whole sub-continent. This is not just a question of the methods used – but rather relates to the assumptions of expanding into already full ecological niches that simply may not have ‘space’ for Lassa to expand into.

Answer: We respectfully disagree with the reviewer. We believe our study examines a question that is of prime interest for Lassa virus surveillance – the potential for virus circulation outside of current endemic areas – and follows a coherent and novel approach by projecting the evolution of the range suitable for Lassa virus transmission across Sub-Saharan Africa in the next decades and evaluating scenarios of virus spread.

It seems that there is a confusion between an actual extension of the Lassa virus range (which is not what we modelled), and a spatial extension of its ecologically suitable area which relates exclusively to the environmental factors suitable for the virus circulation. In our answers above, we detail why we consider that the presence of the main host species (*M. natalensis*) and suitable environmental conditions in some areas of central and eastern Africa may put them at risk of local Lassa virus circulation. We further explore this risk in our phylogeographic simulations, and explicitly repeat throughout the text that we consider that an actual expansion of the virus in those potentially new suitable areas would be a further step.

As mentioned in our response above, the question of a hypothetical competition with arenaviruses circulating locally is out of the scope of our analyses and we thus address that point in the Discussion section of our manuscript.

Minor issues:

L54 – the cases in Ghana have never been confirmed – and indeed one of this manuscript’s authors told me that their further investigations suggested that these cases were not real, so why are they listed here?

Answer: We made our statement based on a report from Dzotsi and colleagues (PMID: 23661832), which describes two cases with a clinical presentation suggestive of Lassa fever. The report specifies that these cases were confirmed by RT-PCR using a published set of primers (PMID: 7883875), which we considered to be a valid confirmation of infection.

L180-182 – the epidemic spread of Ebola in humans, whose spillover related distribution is usually restricted by the endemic dynamics in animal reservoirs bears no relation to the distribution of Lassa, determined by vector dynamics. Further, the translocation of WNV to US from Israel into a novel ecosystem is a qualitatively different consideration to the endemic spread of Lassa regions.

Answer: We agree that our examples of how previous virus introductions into new areas that turned out to be suitable for virus spread bear no relationship to Lassa virus endemic spread. We chose these examples to simply show that viruses can on (unpredictable) occasions travel over long distances, in a manner that does not reflect endemic spread, to settle in new areas. The aim of these examples is to highlight the fact that the presence of suitable areas, even when they are disconnected from the current range of the virus, still presents a risk for viral introduction. This risk is virtually impossible to measure, but still present.

L270 and Table 1. This table is comparing ‘apples and pears’ and is not valuable, but rather I feel is misleading. Spread in epidemic systems in naïve ecological niches (especially human Ebola in West Africa) involves totally different routes etc compared to endemic Lassa transmitting in animal vectors. Similarly many of the rabies transmission rates describe the virus transmitting into naïve populations – something rather different when considering how Lassa needs to compete with other Lassa lineages or other arenaviruses.

Answer: We respectfully disagree with the reviewer. Table 1 compares the exact same metric across different viral diffusion scenarios with the sole purpose of helping the reader get a sense of the speeds that have been measured for other zoonotic viruses because they may not be familiar with the metric we use (weighted lineage dispersal velocity). We fully agree that it would be misleading to attribute the differences in velocity observed between virus dispersal histories to a specific aspect of virus transmission, and we never do so. As the Reviewer rightly points out, various aspects (including the host, route of transmission or immunological status of the population) could explain these differences and we fully agree on this.